# MULTI-LAYER DIFFUSION STRATEGY FOR MULTI-IP INTERACTION-AWARE HUMAN ERASING

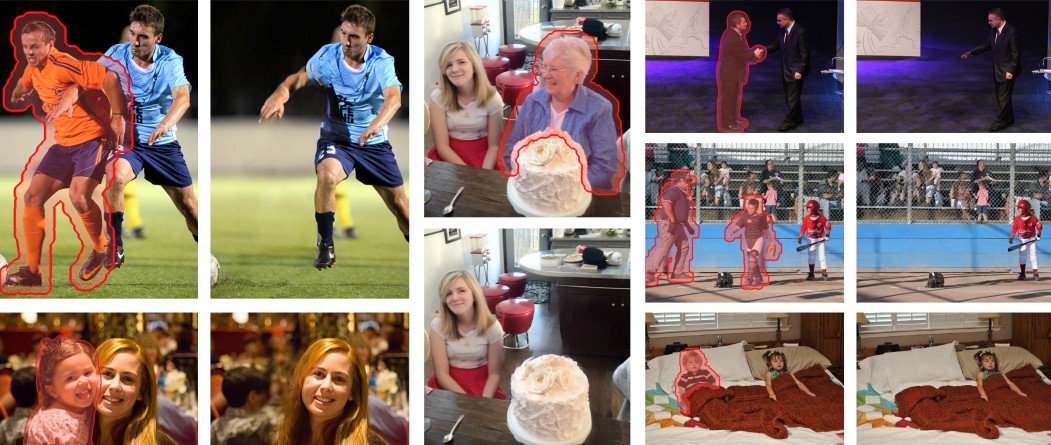

(a) Human-Human Occlusion     (b) Human-Object Entanglement     (c) Human-Background Interference

Figure 1: The figure exhibits MILD (**our method**) can robustly handle three critical challenges in human erasing: *Human–Human Occlusion*, *Human–Object Entanglement*, and *Human–Background interferences* and can achieve clean and artifact-free removal results across diverse scenarios.

## ABSTRACT

Recent years have witnessed the success of diffusion models in image customization tasks. However, existing mask-guided human erasing methods still struggle in complex scenarios such as *human–human occlusion*, *human–object entanglement*, and *human–background interference*, mainly due to the lack of large-scale multi-instance datasets and effective spatial decoupling to separate foreground from background. To bridge these gaps, we curate the *MILD dataset* capturing diverse poses, occlusions, and complex multi-instance interactions. We then define the *Cross-Domain Attention Gap (CAG)*, an attention-gap metric to quantify semantic leakage. On top of these, we propose *Multi-Layer Diffusion (MILD)*, which decomposes the generation process into independent denoising pathways, enabling separate reconstruction of each foreground instance and the background. To enhance human-centric understanding, we introduce *Human Morphology Guidance*, a plug-and-play module that incorporates pose, parsing, and spatial relationships into the diffusion process to improve structural awareness and restoration quality. Additionally, we present *Spatially-Modulated Attention*, an adaptive mechanism that leverages spatial mask priors to modulate attention across semantic regions, further widening the CAG to effectively minimize boundary artifacts and mitigate semantic leakage. Experiments show that MILD significantly outperforms existing methods. Datasets and code are publicly available at: `project page`.

## 1 INTRODUCTION

Object removal aims to eliminate user-specified regions and synthesize visually coherent, context-consistent backgrounds (Jampani et al., 2021; Xu et al., 2023). This capability supports a wide range of applications, including photo editing, privacy preservation, and interactive content creation (Yu

et al., 2018; Liu et al., 2018). However, achieving both thorough removal and faithful background restoration remains a challenging task, especially in scenarios involving multiple interacting instances and dense occlusions.

Early non-parametric methods (Criminisi et al., 2004; Barnes et al., 2009) repainted erased regions by copying patches from visible areas, but often produce repetitive textures and fail in complex scenes. Afterwards, GAN-based models (Pathak et al., 2016; Nazeri et al., 2019) introduce adversarial learning to enhance visual realism, yet suffer from global inconsistency and instability under large masks. Recently, diffusion-based models (Song et al., 2021; Avrahami et al., 2023; Yang et al., 2023a) have achieved competitive performance by leveraging large-scale pretrained generative priors (Rombach et al., 2022b; Ekin et al., 2024). These models (Zhuang et al., 2024; Yildirim et al., 2023) excel in generating diverse content, but remain limited in eliminating objects in complex human-centric multi-instance scenarios.

For example, in Figure 2, standard LDM (Inst-Inpaint)always fails to remove the target cleanly and instead hallucinates a new human-like figure at the masked region. Notably, the generated content retains dominant visual hints, such as "human", implying that semantic features from the removed subject have leaked into the generation process.

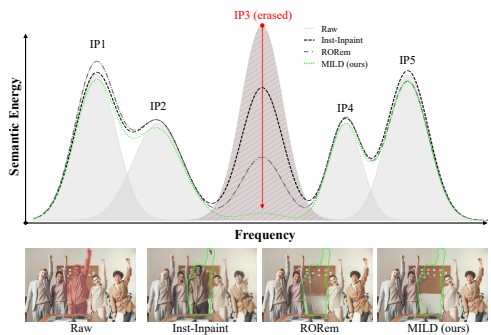

Figure 2: Peak signal of semantic leakage.

These limitations reveal a fundamental challenge: *most existing approaches treat object removal as a unified generation task without explicitly disentangling the generation paths of different foreground instances*. This framework often leads to semantic interference and content entanglement, particularly in multi-interactive-person (Multi-IP) or object-dense scenes, where features from one instance may permeate into others, degrading output quality and controllability.

To address this challenge, we propose MILD (Multi-Layer Diffusion) strategy, a theoretically grounded framework for human-centric object removal. We first theoretically analyze the source of semantic leakage by introducing the Cross-Domain Attention Gap (CAG) to quantify attention flow discrepancies between foreground and background regions. Theoretical analysis demonstrates that enlarging CAG significantly reduces semantic leakage, inspiring our core design principle: *maximizing the attention gap through architectural separation*.

Guided by this theory, MILD reformulates human erasing as a layered diffusion process, establishing independent diffusion paths for each target instance and the background to achieve instance-level disentanglement. Specifically, MILD employs a shared UNet backbone with domain-specific LoRA adapters, ensuring scalability while maintaining attention domain separation between foreground and background. To enhance instance awareness in complex scenarios, we introduce Human Morphology Guidance (HMG), which injects human pose and parsing priors to preserve structural integrity under occlusion. Furthermore, the Spatially-Modulated Attention (SMA) module applies mask-conditioned constraints to attention logits, further expanding CAG and achieving exponential leakage suppression. This design not only enables more thorough instance removal but also produces flexibly recomposable layered outputs, supporting diverse scene editing operations.

To advance research in this field, we release a high-quality human-erasing dataset with diverse scenes, challenging poses, occlusions, and interactions. It provides paired images with precise masks and realistic backgrounds for rigorous real-world evaluation of human-removal models.

Experimental results demonstrate that our MILD significantly outperforms existing methods on complex human removal tasks with superior visual realism, semantic consistency, and perceptual quality, confirming its state-of-the-art performance.

## 2 RELATED WORK

We introduce the recent works concentrated in image inpainting (see Appendix C), object Removal (see Sec. 2) and Layered Generation (see Appendix C).

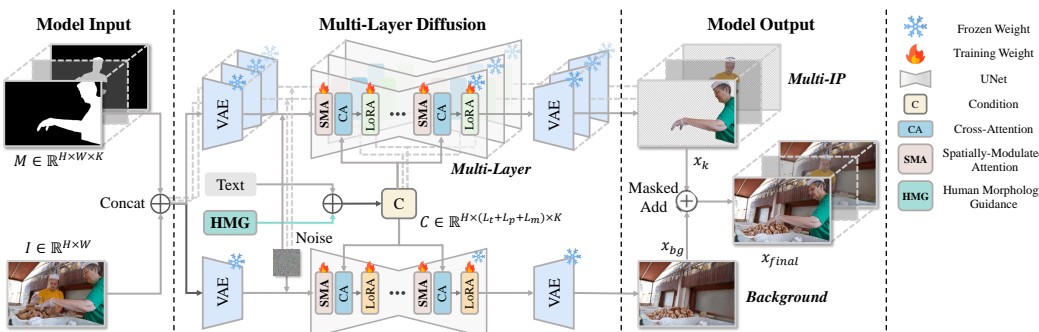

Figure 3: Overview of MILD. Given an input image and a set of target masks, the proposed *Multi-Layer Diffusion (MILD)* strategy performs human erasing by generating disentangled foreground layers and a background layer. A shared UNet with Layered LoRA enables efficient denoising across all branches, producing composable outputs. Spatially-Modulated Attention (SMA) injects adaptive biases to suppress semantic leakage across masked regions. Cross-attention is conditioned on the text prompt and Human Morphology Guidance (HMG) to guide instance-aware generation.

**Object Removal** Object removal, as a task derived from image inpainting, aims to eliminate user-specified content while preserving scene fidelity. Early approaches (Suvorov et al., 2022) achieved reasonable results by adapting general-purpose generative models with task-specific (Jain et al., 2023; Yi et al., 2020b) guidance, but they struggled to capture complex textures and scene structures. Recent works (Hu et al., 2022; Kingma & Welling, 2014; Ronneberger et al., 2015; Rombach et al., 2022a; Ho et al., 2020) have explored a range of strategies to improve controllability and fidelity (Winter et al., 2024). Training-based (Yang et al., 2023b; Jiang et al., 2025; Liu et al., 2025; Li et al., 2025) approaches fine-tune diffusion models using supervision such as masks, prompts, or parameter-efficient modules. Training-free (Sun et al., 2025; Han et al., 2024; Jia et al., 2024; Li et al., 2024a) approaches, in contrast, manipulate internal representations (e.g., attention maps or embeddings) during inference to enable flexible editing. In parallel, different forms of user interaction have been introduced—ranging from text-prompted editing (Bar-Tal et al., 2022; Zhao et al., 2021; Geng et al., 2024) to mask-guided (Li et al., 2022; Köhler et al., 2014; Xie et al., 2023a;b) generation—to support more intuitive control. Despite these advances, we found that most existing methods lack instance-level distinction, generating a single prediction conditioned on all masked targets or prompts. This entangled treatment limits the compositional flexibility and often leads to interference between each instance and its surrounding region, especially when it comes to complex human-oriented interaction scenes.

## 3 METHOD

**Overview** Existing inpainting methods often fail to entirely remove an instance, leaving residual artifacts or introducing semantic leakage in complex scenes. To address these phenomenon, we define *Cross-Domain Attention Gap (CAG)* to quantify the semantic leakage. On the top of our theoritical analysis, we introduce *Multi-Layer Diffusion (MILD)* (Fig. 3), a diffusion-based strategy that reformulates human erasing as a layered diffusion process. MILD ultilizes low-rank adaptation modules as its backbone (Sec. 3.2) to produce disentangled per-instance and background layers. On top of this, *Human Morphology Guidance* (Sec. 3.3) injects pose, parsing, and spatial priors to enhance instance-aware understanding, while *Spatially-Modulated Attention* (Sec. 3.4) suppresses semantic leakage across spatially separated regions using mask priors. Together, these components enable precise, artifact-reduced erasing in scenes with occlusions and entanglements.

### 3.1 THEORETICAL MOTIVATION: DISENTANGLEMENT VIA ATTENTION GAP

Traditional inpainting methods often produce residual artifacts and semantic confusion in multi-instance scenes, which attributing to the lack of the fundamental decoupling between foreground and background in unified generation processes. To address this issue, we first establish a theoretical framework for quantifying semantic leakage.

We introduce the *Cross-Domain Attention Gap (CAG)* as a measure of attention flow discrepancy between foreground and background regions. Our analysis reveals that semantic leakage correlates directly with the magnitude of this gap:

$$\gamma := \inf_{i \in B} \left[ \log \sum_{j \in B} e^{A_{ij}} - \log \sum_{j \in F} e^{A_{ij}} \right], \tag{1}$$

where $A_{ij}$ represents attention scores, and $B$ and $F$ denote background and foreground regions.

Intuitively, $\gamma$ characterizes the relative amount of attention assigned to background versus foreground positions for each background token. A large positive $\gamma$ implies that background tokens predominantly attend to background content, thereby suppressing the influence of foreground semantics in the reconstructed background, whereas a small or negative $\gamma$ indicates that some background locations allocate non-negligible attention to humans, making semantic leakage more likely.

Theorem 1 (proof in Appendix F.3) establishes that a larger $\gamma$ leads to markedly reduced semantic leakage. This theoretical insight motivates our core design principle: *maximizing the attention gap through architectural separation*.

## 3.2 MILD BACKBONE: MULTI-LAYER DISENTANGLEMENT

Building on the theoretical foundation above, we propose *Multi-Layer Diffusion (MILD)*—a diffusion-based strategy that reformulates human erasing as a layered diffusion process. Rather than relying on a unified single-layer pipeline that inherently couples instances, MILD implements the separation principle derived from our CAG analysis.

Given an input image and $N$ target masks $\{M_k\}_{k=1}^N$, MILD produces $N$ foreground layers and a clean background layer. Each foreground branch reconstructs its corresponding subject, while the background branch synthesizes the unobstructed environment. This architectural separation directly implements the attention gap maximization suggested by our theoretical analysis.

To ensure scalability while maintaining separation, we employ a shared UNet with domain-specific LoRA adapters. The parameterization follows our theoretical conditioning requirements:

$$\Delta W_p^{(b)} = \alpha_p^{(b)} B_p^{(b)} A_p^{(b)}, \quad p \in \{Q, K, V\}, \quad b \in \{\text{fg}, \text{bg}\},$$

where foreground and background branches maintain independent adapter sets $(A_p^{(b)}, B_p^{(b)})$, enforcing the attention domain separation identified as crucial for reducing semantic leakage.

The corollaries in Appendix F.4 and F.5 shows that the multi-layer parameterization is better conditioned to train while being no worse in fit than a comparable single-layer model.

During inference, MILD produces composable outputs $\{\text{Layer}_1, \ldots, \text{Layer}_N, \text{Background}\}$ that can be flexibly recombined:

$$x_{\text{final}} = \left( \sum_{k \notin \mathcal{R}} M_k \odot x_k \right) + \left( \left(1 - \sum_{k \notin \mathcal{R}} M_k \right) \odot x_{\text{bg}} \right), \tag{2}$$

where $\mathcal{R}$ denotes instances to remove. This compositional approach directly embodies the decoupling benefits predicted by our CAG theory.

## 3.3 HUMAN MORPHOLOGY GUIDANCE

Although the Layered LoRA diffusion backbone establishes the architectural foundation for instance disentanglement, complex scenarios involving spatial overlap and occlusion require enhanced instance-awareness to maintain structural fidelity. To handle such cases, we introduce *Human Morphology Guidance* (HMG) to reinforce instance identity under challenging conditions by injecting human-centric priors into the denoising process. HMG operates through two complementary guidance mechanisms: pose and parsing guidance and spatial context modeling. We use the same off-the-shelf estimators (Cao et al., 2017; Li et al., 2020b) at training and inference pipeline and these cues are optional. For pose and parsing guidance, we extract pose keypoints and semantic parsing maps, which are encoded through shared convolutional stacks and linearly projected into compact

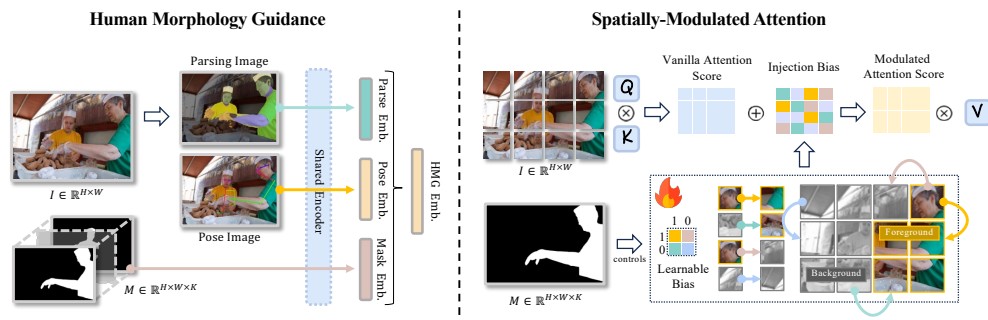

Figure 4: (a) HMG extracts fine-grained human priors from pose and parsing maps using two dedicated encoders. These features, combined with a spatial mask prior, form a unified latent representation that enriches the denoising process with human-centric and spatial cues, improving fidelity and identity consistency. (b) Illustration of the proposed *Spatially-Modulated Attention (SMA)* mechanism. For each query-key pair $(i, j)$, an spatial bias $\alpha_{st}$ is applied to the vanilla attention score $A_{ij}$, where $s = m_i$ and $t = m_j$ indicate their foreground/background status.

morphological embeddings. These features are concatenated with the target instance's mask embedding, providing explicit structural priors that help maintain body coherence under occlusion. To further enhance spatial reasoning, we add a lightweight mask-based context that explicitly models neighboring instances. For each instance $k$, we construct $M^{(k)} = \mathrm{clamp}\left(\sum_{j \neq k} M_j, 0, 1\right)$, which aggregates all other instances into a single binary context. This explicit spatial modeling enhances localization accuracy and reduces interference between adjacent instances. As detailed in the algorithmic appendix (Appendix D), HMG seamlessly integrates into the MILD backbone by conditioning both foreground and background branches, with morphology features optionally incorporated during training and inference. The synergy between HMG's explicit morphological guidance and our theoretical CAG framework yields significant benefits: explicit structural priors reduce the need for cross-instance attention, thereby amplifying the attention gap $\gamma$ identified in Section 3.1.

## 3.4 SPATIALLY-MODULATED ATTENTION

Even though MILD backbone and HMG works to meet our decoupling design, we still observe residual semantic leakage due to limitations in the CAG under vanilla attention mechanisms.

This motivates modulating the vanilla attention logits to enlarge the gap(*i.e.*, $\tilde{\gamma} \geq \gamma$), thereby tightening the leakage bounds. We then propose *Spatially-Modulated Attention* (SMA), a lightweight module that enforces spatial constraints based on mask boundaries to prevent the model from indiscriminately using the entire context to synthesize background, as shown in Fig. 4.

In standard self-attention, logits are $A_{ij} = \frac{Q_i K_j^\top}{\sqrt{d_k}}$. To control attention flow across semantically separated regions, SMA adds a mask-conditioned bias using binary region labels $m_i, m_j \in \{0, 1\}$ (1 = foreground, 0 = background):

$$\tilde{A}_{ij} = A_{ij} + \sum_{s,t \in \{0,1\}} \alpha_{st} \mathbb{1}[m_i = s, m_j = t], \tag{3}$$

where $\alpha_{st}$ are four learnable scalars, maintained per self-attention block and shared across its attention heads. The indicator $\mathbb{1}[m_i = s, m_j = t]$ equals 1 when token $i$ lies in region $s$ and token $j$ in region $t$, and 0 otherwise.

We initialize all biases at zero to preserve vanilla attention at the start and enable adaptive spatial modulation during training. To encourage suppression of cross-region interactions, we enforce hard constraints on the bias terms: $\alpha_{10}, \alpha_{01} \leq 0$ (penalizing cross-domain attention) and $\alpha_{11}, \alpha_{00} \geq 0$ (reinforcing within-domain attention). These constraints are implemented as hard boundaries during optimization, ensuring consistent suppression of cross-region interactions throughout training.

Figure 5: Qualitative results produced by MILD (ours) and other methods in real-world scenes. The masked regions (in red) and the corresponding removal results (in green) are highlighted.

This design provides explicit, mask-driven control over attention flow without altering the core attention mechanism. Deploying SMA mechanism widens the CAG, cuts leakage exponentially (Theorem 2), and improves training conditioning and stability at minimal cost.

## 3.5 OPTIMIZATION OBJECTIVE

During training, each foreground branch learns to reconstruct its assigned region while the background branch predicts the noise component in the diffusion process. Following standard diffusion model formulation, given a clean image $x_0$ and a randomly sampled timestep $t$, we obtain the noisy input $x_t = \sqrt{\bar{\alpha}_t}x_0 + \sqrt{1 - \bar{\alpha}_t}\epsilon$ where $\epsilon \sim \mathcal{N}(0, I)$ is the target noise. Each branch is trained to predict this target noise within its respective domain:

$$\mathcal{L}_k = \left\| M_k \odot \left( \epsilon - \epsilon_k(x_t, t) \right) \right\|_2^2, \quad \mathcal{L}_{\text{bg}} = \left\| \epsilon - \epsilon_{\text{bg}}(x_t, t) \right\|_2^2. \quad (4)$$

where $\epsilon_k(x_t, t)$ and $\epsilon_{\text{bg}}(x_t, t)$ denote the noise predictions from the $k$-th foreground branch and background branch respectively, conditioned on the noisy input $x_t$ and timestep $t$.

The total loss combines foreground and background terms with a training-step warm-up:

$$\mathcal{L}_{\text{total}} = \lambda_s \sum_{k=1}^{N} \mathcal{L}_k(\theta_{\text{fg}}) + \mathcal{L}_{\text{bg}}(\theta_{\text{fg}}, \theta_{\text{bg}}), \quad \lambda_s = \begin{cases} 0, & s < s_0, \\ \lambda \cdot \dfrac{s - s_0}{s_1 - s_0}, & s_0 \leq s \leq s_1, \\ \lambda, & s > s_1, \end{cases} \quad (5)$$

where $s$ is the training iteration index, and $\theta_{\text{fg}}$ and $\theta_{\text{bg}}$ refer to the foreground and background parameters respectively. We adopt a staged strategy, first optimizing the background branch before ramping in foreground supervision.

## 4 EXPERIMENTS

### 4.1 EXPERIMENTAL SETUP

**Datasets** We conduct comprehensive evaluations on two datasets to assess both task-specific performance and cross-domain generalization: 1) *Our MILD Dataset*: We construct a high-quality dataset specifically tailored for multi-IP erasure. The dataset comprises $10,000$ image pairs featuring complex human interactions, diverse body poses and challenging occlusions. The detailed information is introduced in Appendix A. 2) *OpenImages Dataset*: To evaluate generalization beyond human-centric scenarios, we randomly sample 1,000 images from OpenImages V5 (Kuznetsova et al., 2020), covering diverse object categories and photographic conditions.

| Method | MILD Dataset | | | | | OpenImages Dataset | | | | |
|---|---|---|---|---|---|---|---|---|---|---|
| | FID↓ | LPIPS↓ | DINO↑ | CLIP↑ | PSNR↑ | FID↓ | LPIPS↓ | DINO↑ | CLIP↑ | PSNR↑ |
| SDXL Inpaint | 53.45 | 0.242 | 0.8398 | 0.8331 | 18.22 | 26.22 | 0.1346 | 0.8741 | 0.8751 | 21.88 |
| PowerPaint | 48.04 | 0.207 | 0.8901 | 0.8668 | 20.27 | 10.56 | 0.0488 | 0.9649 | 0.9678 | 31.04 |
| Inst-Inpaint | – | – | – | – | – | 11.42 | 0.4100 | 0.7400 | 0.8207 | 23.75 |
| LaMa | 35.02 | 0.164 | 0.8909 | 0.8872 | 24.13 | **10.38** | **0.0470** | 0.9192 | 0.9293 | 31.83 |
| CLIPAway | 48.86 | 0.230 | 0.8647 | 0.8605 | 19.49 | 22.73 | 0.1283 | 0.8973 | 0.8985 | 23.59 |
| RoRem | 40.41 | 0.196 | 0.8888 | 0.8810 | 22.35 | 18.23 | 0.0982 | 0.9095 | 0.9148 | 26.59 |
| **MILD (Ours)** | **24.20** | **0.093** | **0.9703** | **0.9499** | **26.24** | 17.86 | 0.0668 | **0.9829** | **0.9700** | **32.14** |

Table 1: Quantitative comparison of MILD and other methods on MILD and OpenImages datasets. Best results are **bold**, second best are underlined. Inst-Inpaint is excluded from evaluation on MILD dataset as it lacks mask guidance, making it unsuitable for precise localization in multi-IP scenarios.

**Baseline Comparisons** We compare against a diverse set of state-of-the-art object removal methods, including SDXL Inpainting (Podell et al., 2023), LaMa (Suvorov et al., 2022), Power-Paint (Zhuang et al., 2024), Inst-Inpaint (Yildirim et al., 2023), CLIPAway (Ekin et al., 2024), and RoRem (Li et al., 2025), covering paradigms from mask-guided inpainting to instruction-based and vision-language-guided editing. To ensure a fair comparison, we adopt the official open-source implementations and reproduce results under a unified evaluation pipeline. All models are either used as publicly released or retrained with default settings on our datasets to ensure fair comparison.

**Evaluation Metrics** We adopt five complementary metrics to evaluate object removal quality from perceptual, semantic, and low-level fidelity perspectives. FID (Heusel et al., 2017) evaluates global visual realism by comparing feature distributions of generated and real images, while LPIPS (Zhang et al., 2018) focuses on local perceptual similarity based on deep features. DINO cosine similarity (Caron et al., 2021) and CLIP Image Similarity (Radford et al., 2021) measure semantic consistency by computing cosine similarity between the embeddings of generated and ground-truth images, capturing structural and high-level semantic alignment, respectively. Finally, PSNR (Hore & Ziou, 2010) quantifies pixel-level fidelity within the removed regions.

## 4.2 QUANTITATIVE AND QUALITATIVE COMPARISONS

Table 1 summarizes the quantitative performance of various object removal methods on the MILD and OpenImages datasets. Our proposed MILD strategy consistently outperforms all baselines on the more challenging MILD dataset and remains highly competitive on OpenImages. Compared to the SDXL Inpainting baseline, MILD reduces FID and LPIPS by 54.53% and 61.57% respectively, reflecting significant improvement in visual realism and perceptual quality. It also achieves higher DINO, CLIP, and PSNR scores, demonstrating enhanced semantic consistency and pixel-level fidelity through its spatially-aware layered diffusion design.

While LaMa performs well on FID and LPIPS in relatively simple scenes due to its smooth outputs, it underperforms in semantic metrics like DINO and CLIP, revealing limitations in understanding complex content. In contrast, MILD delivers superior results across all dimensions, particularly in complex human interactive scenes.

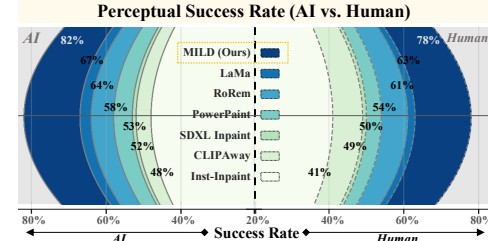

Figure 6: Comparison of AI and human success rates for object removal across different methods on the MILD dataset.

Figure 5 exhibits qualitative comparisons in challenging multi-IP scenarios. When the masked human occupies a large image region (**Row 1**), MILD generates semantically coherent and visually realistic content outperforming baselines. In scenes with complex human interactions (**Row 3**), it produces cleaner boundaries and fewer re-drawings. Compared to existing methods, MILD effectively reduces hallucinations, avoids over-smoothing and blurring, and reconstructs sharper details in occluded regions. Notably, while LaMa performs well in simpler cases, it struggles with complex semantics and often introduces unrealistic blending (**Row 2**), which consistent with our quantitative results. These results demonstrate the effectiveness of MILD's layered diffusion strategy and spatially guided conditioning.

Figure 7: Qualitative ablation comparison of our method.

| Backbone | HMG | SMA | Full | FID↓ | Δ | LPIPS↓ | Δ | DINO↑ | Δ | CLIP↑ | Δ | PSNR↑ | Δ |
|---|---|---|---|---|---|---|---|---|---|---|---|---|---|
| ✗ | ✗ | ✗ | ✗ | 53.45 | +29.25 | 0.242 | +0.149 | 0.8398 | -0.1305 | 0.8331 | -0.1168 | 18.22 | -8.02 |
| ✓ | ✗ | ✗ | ✗ | 29.02 | +4.82 | 0.142 | +0.049 | 0.9110 | -0.0593 | 0.9025 | -0.0474 | 23.45 | -2.79 |
| ✓ | ✓ | ✗ | ✗ | 26.10 | +1.90 | 0.120 | +0.027 | 0.9440 | -0.0263 | 0.9312 | -0.0187 | 24.67 | -1.57 |
| ✓ | ✗ | ✓ | ✗ | 25.10 | +0.90 | 0.105 | +0.012 | 0.9380 | -0.0323 | 0.9227 | -0.0272 | 25.21 | -1.03 |
| ✓ | ✓ | ✓ | ✓ | **24.20** | – | **0.093** | – | **0.9703** | – | **0.9499** | – | **26.24** | – |

Table 2: Ablation study on the MILD dataset comparing the SDXL baseline, the model with the proposed MILD backbone, and the incremental addition of HMG and SMA.

While traditional quantitative metrics provide valuable evaluations of model performance, they often fail to capture the direct naturalness and visual quality as perceived by human observers. To address this limitation, we conduct a comprehensive perceptual evaluation from both AI and human perspectives. The evaluation of both assessments are summarized in Figure 6, and the detailed experiment setting and the full table is shown in Appendix G.6. Both AI and human evaluations consistently demonstrate that MILD achieves superior perceptual quality compared to baseline methods. Our method (MILD) achieves the highest success rate for both AI (82%) and human raters (78%), indicating consistent perceptual superiority. This comprehensive perceptual assessment validates the effectiveness of our approach in producing human-preferred removal results.

In summary, MILD achieves state-of-the-art performance on complex human interaction scenes, while also demonstrating competitive generalization to broader domains. It effectively balances visual realism, semantic alignment and perceptual quality across diverse human erasing scenarios.

## 4.3 ABLATION STUDY

To evaluate the impact of the components in our multi-layer diffusion strategy, we conduct a step-by-step ablation study with both the quantitative and qualitative results in Table 4.3 and Figure 7. We progressively modify the baseline through four main experiments. We first incorporate the MILD Backbone into the SDXL inpainting baseline. Then the HMG and SMA modules are separately added into the structure. Finally, we evaluate the full MILD model, which integrates the MILD Backbone along with both HMG and SMA modules.

The single layer baseline performs poorly(row 1) on the MILD dataset, often produces incomplete removals or hallucinated content when handling large or occluded subjects. Introducing MILD Backbone significantly improves both perceptual and semantic quality by disentangling generation into background and instance-specific foreground branches. This reduces the repainting phenomenon with 45.71% improvement in FID metric. Building on this backbone, HMG and SMA provide complementary enhancements. As shown in Figure 7, HMG helps to preserve body structure (row 2) and relieve the residual artifacts along object boundaries(row 1). Meanwhile, SMA focuses on refining the attention flow, effectively suppressing the unnatural mask bleeding and background leakage, resulting in smoother transitions and more coherent backgrounds. The full MILD model combines these advantages, achieving the best overall performance across all metrics. These results validate the non-redundant and effectiveness of our modular design.

Figure 8: Illustration of MILD's compositional flexibility.

# 5 DISCUSSION

## 5.1 WHY DOES MILD SUPPRESS FREQUENCY PEAKS MORE EFFECTIVELY?

To understand why MILD achieves superior suppression of residual frequency peaks compared to conventional methods, as shown in Fig. 2, we analyze semantic leakage as a form of cross-domain attention transfer from foreground to background. Let $A = QK^\top/\sqrt{d_k}$ and $P = \mathrm{softmax}(A)$, and let $M \in \{0, 1\}^{N \times N}$ mark foreground–background pairs. The operator $\mathcal{T}(P) = P \odot M$ captures the background-side mass originating from the foreground and correlates with the erased-instance peak in frequency plots. Spatially-Modulated Attention applies a negative logit bias $-\beta M$ to cross-domain pairs, yielding $\tilde{A} = A - \beta M$ and $\tilde{P} = \mathrm{softmax}(\tilde{A})$. By a row-mass argument (see App. F.6),

$$\left\|\mathcal{T}(\tilde{P})\right\|_\infty \leq e^{-\beta} \left\|\mathcal{T}(P)\right\|_\infty. \tag{6}$$

In parallel, a Jacobian-based estimate provides an additive Lipschitz bound in $\ell_2$ for small $\beta$ (App. F.6). Since the cross-domain map in MILD composes $LT$ attention blocks with per-block biases $\{\beta_{\ell,t}\}$, submultiplicativity gives

$$\left\|\mathcal{T}_{\mathrm{MILD}}\right\|_\infty \leq \exp\Big(-\sum_{\ell=1}^{L}\sum_{t=1}^{T}\beta_{\ell,t}\Big) \left\|\mathcal{T}(P_{\mathrm{base}})\right\|_\infty = e^{-\bar{\beta}LT} \left\|\mathcal{T}(P_{\mathrm{base}})\right\|_\infty, \tag{7}$$

where $\bar{\beta} = \frac{1}{LT}\sum_{\ell,t}\beta_{\ell,t}$. Eq. 6 and Eq. 7 explain the plots: the erased-instance peak mass admits an upper bound decaying approximately as $e^{-\bar{\beta}LT}$, i.e., linear in depth and steps on a log scale.

## 5.2 HOW DOES MILD ENABLE FLEXIBLE SCENE RECOMPOSITION?

Beyond achieving high-quality foreground removal and background restoration, MILD's layered diffusion design introduces a key advantage: the ability to flexibly recompose scenes using instance-specific foreground layers and a clean background. As shown in Figure 8, MILD generates disentangled outputs for each target, enabling selective combination to construct customized scenes. This allows users to separate specific individuals (e.g., only IP1), remove arbitrary targets (e.g., w/o IP3), or reconstruct any subset of the original scene. Such instance-aware recomposition offers fine-grained control over content manipulation, making MILD well-suited for interactive editing and scene simulation tasks. More details are shown in Appendix G.7.

# 6 CONCLUSION

We present *Multi-Instance Layered Diffusion*, a diffusion-based strategy reformulating inpainting as a layered, decoupled diffusion process for precise human erasing, supported by our curated, high-quality *MILD dataset*. On top of our theoretical analysis based on CAG, MILD leverages multi-layer structure along with HMG and SMA to provide instance-aware structure understanding and brings spatially-conditioned attention control, which effectively reducing artifacts and improving reconstruction fidelity. Extensive experiments show that MILD outperforms existing inpainting baselines, achieving better semantic consistency, fewer artifacts, and more coherent scene reconstructions. MILD provides a promising direction for scaling precise diffusion-based inpainting to complex multi-object and open-domain removal tasks.

## 7 ETHICS STATEMENT

When gathering our dataset, we ensure that raters are compensated. We also run safety filters over the generated images before giving them to the raters. This work is a step towards better evaluation of text-to-image models which are known to hallucinate. It gives tools to others developers and practitioners to properly understand and evaluate T2I models in the future.

## 8 REPRODUCIBILITY STATEMENT

Our project page, along with our code and dataset is available at: `project page`. We give extensive details of our experimental setup in Section 4.1 and Appendix G.1. For human annotation, we visualise the templates used and give extensive detail on how these raw ratings are aggregated in G.6. For our dataset , the details are stated in the Appendix A, and the sample data pairs are submitted as supplementary materials. For metrics, we give full details of the baselines in Section 4 and the evaluation pipeline will be released along with code upon acceptance. For model design, we provide the core algorithm in Appendix D, and the code will be released upon acceptance. For theoretical results, clear explanations of any assumptions and a complete proof of the claims are included in the Appendix E and Appendix F.

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

## A  DATASET AND CODE AVAILABILITY

**Dataset Construction.**  Our dataset consists of high-quality images sourced from two main origins: (1) a carefully curated subset of the OpenImages V5 dataset (Kuznetsova et al., 2020), and (2) publicly available web images, collected under fair use for academic research purposes. All images are manually verified to ensure diversity in scenes, human poses, occlusion patterns, and background complexity.

Each image is annotated with pixel-level, instance-specific human masks. In addition, we provide human-selected ground-truth completions to enable supervised evaluation of object removal performance. Figure 9 presents representative examples from the dataset. The full dataset will be publicly released on our project page upon acceptance, and the sample dataset is provided in our supplementary materials.

**Code and Model Availability.**  To facilitate future research and ensure reproducibility, we have created an anonymous project page containing representative examples and the paper abstract. Our core codebase and pre-trained models will be updated upon acceptance.

## B  THE USE OF LARGE LANGUAGE MODELS (LLMS)

Large language models (LLMs) were used for grammar and formatting checks, as well as for translation assistance.

## C  MORE RELATED WORKS

**Image Inpainting**  Image inpainting aims to restore missing or occluded regions in a visually coherent and semantically consistent manner in various scenarios (Sargsyan et al., 2023; Zhou et al., 2020; Wang et al., 2024; Yoon & Cho, 2024). Traditional methods relied on patch propagation (Hays & Efros, 2007; Ding et al., 2019), transferring low-level textures from visible regions but failing in semantically complex scenes. With the rise of deep learning, the encoder-decoder (Zhou et al., 2023; Cao et al., 2023; Altinel et al., 2018; Li et al., 2020a; Yi et al., 2020a) and GAN-based architectures (Iizuka et al., 2017; Yu et al., 2018; Zeng et al., 2019) have significantly improved structural plausibility and perceptual quality, further enhanced by edge guidance (Nazeri et al., 2019) and attention mechanisms (Yu et al., 2018). Diffusion-based methods (Lugmayr et al., 2022; Saharia et al., 2022) have recently achieved strong generation performance by operating in latent spaces guided by powerful pretrained priors, but they often lack precise spatial control or instance-level disentanglement. Methods like SDXL Inpainting (Podell et al., 2023) introduce mask-based conditioning for edit localization, but treating all masked regions as single entity prevents instance-specific control. Meanwhile, recent efforts incorporate large language models (LLMs) to enable instruction-based editing (Brooks et al., 2023; Huang et al., 2024; Li et al., 2024b; Sheynin et al., 2024; Li et al., 2024c), yet they struggle to localize individual instances in complex scenes.

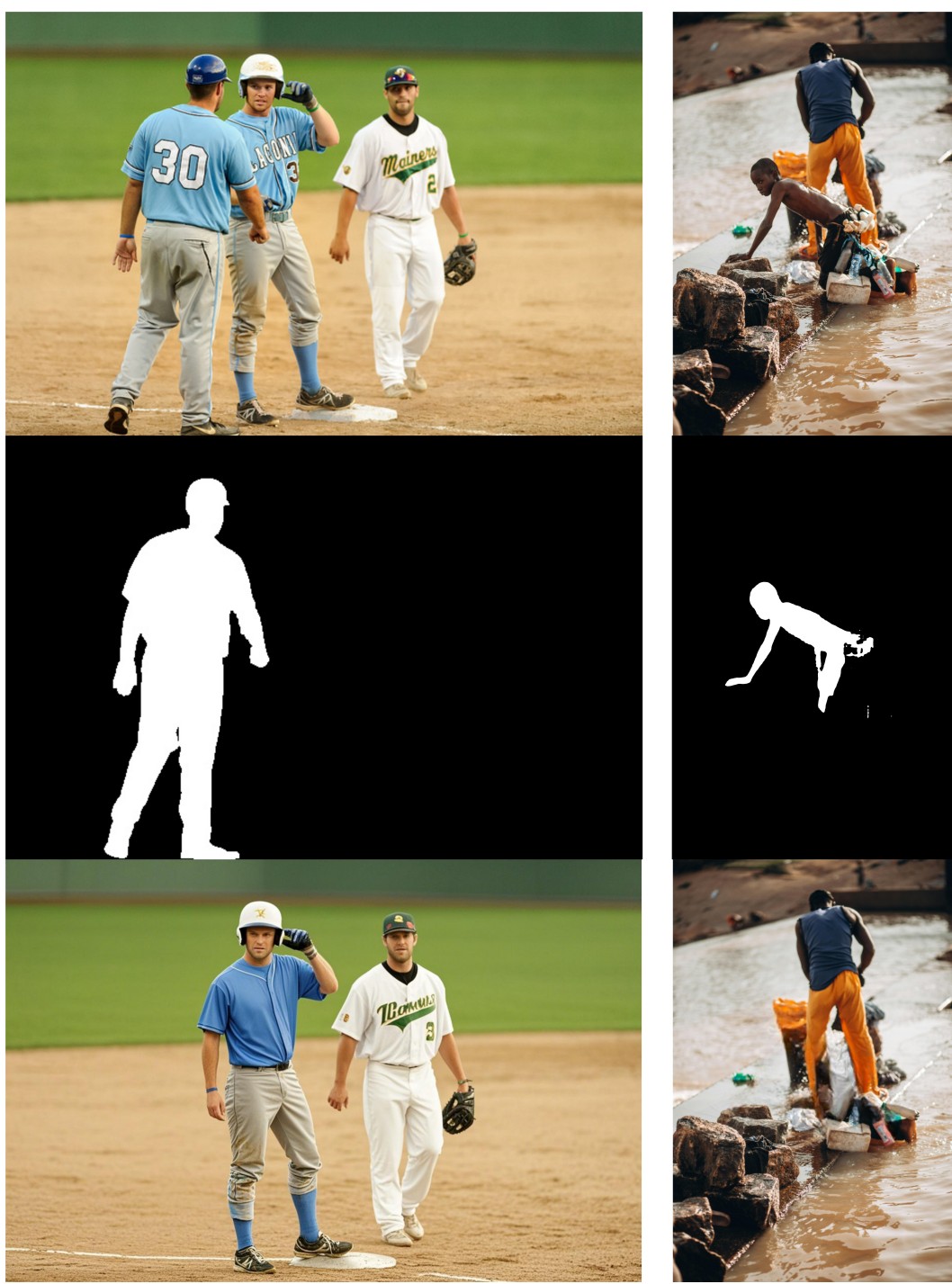

Figure 9: Sample data collected from MILD dataset.

**Layered Generation and Controllability.** Recent works on image generation have explored layered modeling and layout-based generation to enhance object awareness and controllability. Layered modeling works (Jia et al., 2024; Lu et al., 2020) leverage multi-level structures through latent decomposition and scene-layer separation to enable semantically consistent editing. However, they often operate within a unified generation pathway and lack explicit instance disentanglement

---

**Algorithm 1** MILD Training

---

**Require:** Image input: $x$, masks $\{M_k\}_{k=1}^N$
**Require:** Condition inputs: text prompt $c_{\text{text}}$, pose keypoints $P$, parsing maps $S$
1: **// Human Morphology Guidance**
2: $F_{\text{pose}}, F_{\text{parse}} \leftarrow \text{PoseEncoder}(P, S)$
3: $Z_{\text{pose}}, Z_{\text{parse}} \leftarrow \text{LinearProj}(F_{\text{pose}}, F_{\text{parse}})$
4: **for** $k = 1$ to $N$ **do**
5: $\quad M_{\text{others}}^{(k)} \leftarrow \text{Aggregate}(\{M_j\}_{j \neq k})$
6: $\quad Z_{\text{mask}}^{(k)} \leftarrow \text{MaskEncoder}(M_{\text{others}}^{(k)})$
7: $\quad C^{(k)} \leftarrow \text{Concat}(Z_{\text{text}}, Z_{\text{pose}}, Z_{\text{parse}}, Z_{\text{mask}}^{(k)})$
8: **end for**
9: $C_{\text{bg}} \leftarrow \text{Concat}(c_{\text{text}}, T_{\text{pose}}, T_{\text{parse}})$
10: **// Dual LoRA Training**
11: **for** each timestep $t$ and noise $\epsilon$ **do**
12: $\quad z_t \leftarrow \sqrt{\bar{\alpha}_t} z + \sqrt{1 - \bar{\alpha}_t} \epsilon$
13: $\quad$ **for** $k = 1$ to $N$ **do**
14: $\quad\quad \mathcal{L}_k \leftarrow \|M_k \odot (\epsilon - \epsilon_k)\|_2^2$ {Foreground branch}
15: $\quad$ **end for**
16: $\quad \mathcal{L}_{\text{bg}} \leftarrow \|\epsilon - \epsilon_{\text{bg}}\|_2^2$ {Background branch}
17: $\quad \mathcal{L}_{\text{total}} \leftarrow \lambda_t \sum_{k=1}^N \mathcal{L}_k + \mathcal{L}_{\text{bg}}$
18: **end for**

---

or spatial conditioning, making them less suitable for multi-object removal. Prompt- and layout-controllable generation works (Liu et al., 2022; Zheng et al., 2023; Zhangli et al., 2024) aim to synthesize scenes from object-level prompts or layout priors. Although these methods offer flexible conditioning and structural guidance, they generally operate at a holistic level without explicit separation of foreground instances, limiting their capacity to handle occlusions or support fine-grained instance control. In contrast, our approach introduces a mask-aware, multi-branch generation framework that explicitly disentangles each foreground instance and the background into independent pathways. This layered formulation enables precise object removal, instance-level recomposition, and enhanced semantic fidelity through joint supervision.

## D    MORE ALGORITHMIC DETAILS

To ensure reproducibility, we provide additional details of the training pipeline, architectural design, and spatial supervision strategies used in *LayeredUNet*. The model is optimized in stages (Algorithm 2), where a training controller gradually activates different generation branches. The training detail is in Algorithm 1. Each image is encoded into the latent space and perturbed with spatial offsets derived from the union of all masks. To stabilize learning, the background branch is initially protected by freezing its gradients, while foreground branches are progressively optimized with instance-specific conditioning.

Both foreground and background branches share a unified UNet backbone but use independent LoRA adapters. During the forward pass (Algorithm 3), we apply **Spatially-Modulated Attention (SMA)** in all attention layers across both branches, replacing standard self-attention. SMA enforces spatial constraints guided by the input masks, helping to prevent feature leakage and preserve structural consistency.

To further improve boundary quality and compositional integrity, optional modules such as layer exchange and boundary smoothing can be enabled. These help align overlapping regions and reduce visible seams. Additionally, the foreground branch is supervised using a **region-aware loss** that emphasizes both the core masked region and its boundaries via residual and gradient-based penalties, improving local coherence and mask adherence.

---

**Algorithm 2** Training of LayeredUNet with Progressive Dual LoRA

---

**Require:** Input image $x$, mask set $\{M_k\}_{k=1}^N$, prompt $c_{\text{text}}$, optional pose $P$, parsing $S$
1: $z_0 \leftarrow \text{VAE}(x)$
2: $z_0 \leftarrow z_0 + \mathcal{E}_{\text{trans}}(\text{union}(\{M_k\}))$ {Inject spatial offsets}
3: $Z_{\text{pose}}, Z_{\text{parse}} \leftarrow \text{PoseEncoder}(P, S)$
4: $Z_{\text{text}} \leftarrow \text{TextEncoder}(c_{\text{text}})$
5: $Z_{\text{cond}} \leftarrow \text{Concat}(Z_{\text{text}}, Z_{\text{pose}}, Z_{\text{parse}})$
6: **for** each training step $t$ **do**
7: $\quad$ stage $\leftarrow \text{GetTrainingStage}(t)$
8: $\quad z_t \leftarrow \text{AddNoise}(z_0, t)$
9: $\quad$ **for** $k = 1$ to $N$ **do**
10: $\quad\quad M_{\text{others}}^{(k)} \leftarrow \text{Aggregate}(\{M_j\}_{j \neq k})$
11: $\quad\quad Z_{\text{mask}}^{(k)} \leftarrow \text{MaskEncoder}(M_{\text{others}}^{(k)})$
12: $\quad\quad C_k \leftarrow \text{Concat}(Z_{\text{cond}}, Z_{\text{mask}}^{(k)})$
13: $\quad\quad \epsilon_k \leftarrow \text{UNet}(z_t, t, C_k, \text{branch} = \texttt{"fg"}, M_k)$
14: $\quad\quad \mathcal{L}_k \leftarrow \text{RegionLoss}(\epsilon_k, z_0, M_k)$
15: $\quad$ **end for**
16: $\quad \epsilon_{\text{bg}} \leftarrow \text{UNet}(z_t, t, Z_{\text{cond}}, \text{branch} = \texttt{"bg"})$
17: $\quad \mathcal{L}_{\text{bg}} \leftarrow \|\epsilon - \epsilon_{\text{bg}}\|_2^2$
18: $\quad \mathcal{L}_{\text{total}} \leftarrow \lambda_t \sum_k \mathcal{L}_k + \mathcal{L}_{\text{bg}}$
19: $\quad$ **if** stage requires gradient protection **then**
20: $\quad\quad \text{FreezeLoRA}(\texttt{"background"})$
21: $\quad$ **end if**
22: $\quad \text{Backward}(\mathcal{L}_{\text{total}}), \text{UpdateLoRA}()$
23: **end for**

---

**Algorithm 3** Forward Pass of LayeredUNet with SMA

---

**Require:** Latent input $z_t$, timestep $t$, condition $C$, mask $M$, branch $\in \{\texttt{"foreground"}, \texttt{"background"}\}$
1: $z_t \leftarrow z_t + \mathcal{E}_{\text{trans}}(M)$ {Inject spatial offsets}
2: **for** each attention layer in UNet **do**
3: $\quad$ **if** branch = $\texttt{"foreground"}$ **then**
4: $\quad\quad Q, K, V \leftarrow \text{LoRA}_{\text{fg}}(z_t, C)$
5: $\quad\quad \text{attn} \leftarrow \text{SMA}(Q, K, V)$
6: $\quad$ **else**
7: $\quad\quad Q, K, V \leftarrow \text{LoRA}_{\text{bg}}(z_t, C)$
8: $\quad\quad \text{attn} \leftarrow \text{SMA}(Q, K, V)$
9: $\quad$ **end if**
10: $\quad z_t \leftarrow \text{Apply}(z_t, \text{attn})$
11: $\quad$ **if** layer exchange enabled **then**
12: $\quad\quad z_t \leftarrow z_t + \text{LayerExchange}(z_t)$
13: $\quad$ **end if**
14: $\quad z_t \leftarrow z_t + \text{BoundarySmoothing}(z_t, M)$
15: **end for**
16: **return** $\epsilon_{\text{pred}} \leftarrow \text{UNetOutput}(z_t)$

---

# E  THEORETICAL ANALYSIS

## E.1  MULTI-LAYER STRUCTURE

We then analyze the training objective, define the background-favoring margin:

$$\gamma := \inf_{i \in B} \left[ \log \sum_{j \in B} e^{A_{ij}} - \log \sum_{j \in F} e^{A_{ij}} \right]. \tag{8}$$

**Theorem 1** (see proof in Appendix F.3). *Let $H$ be the Hessian of $\mathcal{L}_{total}$ with respect to $(\theta_{\text{fg}}, \theta_{\text{bg}})$. Then*

$$H = \begin{bmatrix} H_{\text{fg}} & C \\ C^\top & H_{\text{bg}} \end{bmatrix}, \tag{9}$$

*and the following bounds hold.*

*If all cross-domain attention logits are set to $-\infty$, then*

$$\|C\|_2 = 0. \tag{10}$$

*Otherwise, define the background-favoring margin $\gamma$ with Eq. 8. Assuming $\gamma \geq 0$, there exists $K > 0$ such that*

$$\|C\|_2 \leq K e^{-\gamma}. \tag{11}$$

Here $C$ is the cross-partial block that measures foreground–background gradient coupling. The constant $K$ aggregates network Lipschitz and loss-smoothness factors, depending only on operator norms and Lipschitz constants of the backbone and on the softmax Jacobian bound.

Let $H_0 := \text{diag}(H_{\text{fg}}, H_{\text{bg}})$ and $m := \lambda_{\min}(H_0)$. We use a scalar weight $\lambda > 0$ in the analysis, and during training a step-dependent schedule $\lambda_s$ may be used. Meanwhile, we use the operator (spectral) norm $\|\cdot\|_2$ and the spectral condition number $\kappa(\cdot)$.

**Corollary 1** (see Appendix F.4 for notation and proof). *Under the assumptions of Theorem 1, if $K e^{-\gamma} < m$, then*

$$\kappa(H_{\text{multi}}) \leq \kappa(H_0) \left(1 + \frac{K}{m} e^{-\gamma}\right). \tag{12}$$

*Moreover, for any single-layer baseline sharing the same diagonal blocks and satisfying $\|\widehat{C}\|_2 \geq \|C\|_2$,*

$$\kappa(H_{\text{multi}}) \leq \kappa(H_{\text{single}}). \tag{13}$$

They provide an *exponentially improving upper bound* on conditioning under increasing margin $\gamma$, with the limit $\kappa(H_0)$ at hard-gating.

**Corollary 2** (see proof in Appendix F.5). *Under a quadratic approximation and a matched $\ell_2$-regularization budget, the multi-layer parameterization attains no larger minimum empirical risk than the single-layer one:*

$$\min_{\theta_{\text{fg}}, \theta_{\text{bg}}} \mathcal{L}_{\text{total}}^{\text{multi}} \leq \min_{\theta} \mathcal{L}_{\text{total}}^{\text{single}}. \tag{14}$$

This follows by viewing the single-layer model as a constrained instance of the multi-layer model with tied parameters and a matched $\ell_2$ budget.

For the multi-layer parameterization, the cross-domain Hessian block satisfies $\|C\|_2 \leq K e^{-\gamma}$ (Theorem 1), yielding a conditioning *upper bound* that approaches the block-diagonal limit as $\gamma$ increases. Meanwhile, the comparative bound to single-layer baselines holds under the norm condition in Corollary 1. Under a matched $\ell_2$ budget, the multi-layer minimum empirical risk is not larger than the single-layer counterpart (Corollary 2).

### E.2 SEMANTIC SUPPRESSION VIA SMA

Formally, treat $X = (X_F, X_B)$ and $Y(u)$ as random variables; if exogenous randomness $Z \perp X$ exists (e.g., diffusion noise), interpret all conditional-independence and CMI statements conditioned on $Z$.

**Theorem 2** (see proof in Appendix F.2). *Assume a deterministic backbone so that functional conditional independences imply zero conditional mutual information. Then the following hold. If all cross-domain logits (i.e., $\alpha_{01}, \alpha_{10}$) are set to $-\infty$, then for any*

$$u \in B^{(-R_{\text{eff}})} := \{ u \in B : \text{dist}(u, F) > R_{\text{eff}} \}, \tag{15}$$

*the background output is functionally independent of the foreground:*

$$I\big(Y(u); X_F \mid X_B\big) \;=\; 0. \tag{16}$$

*If $\gamma > 0$, there exists $C_{\mathrm{net}} > 0$ (depending only on layer operator norms and nonlinearity Lipschitz constants) such that*

$$\big\|\nabla_{\Delta W^{(\mathrm{fg})}} \mathcal{L}_{\mathrm{bg}}\big\| \;\leq\; C_{\mathrm{net}}\, e^{-\gamma}, \tag{17}$$

$$\|g(\xi) - g(\xi')\| \;\leq\; C_{\mathrm{net}}\, e^{-\gamma}\, \|\xi - \xi'\|. \tag{18}$$

Here $g : \mathcal{X}_F \to \mathcal{Y}_B$ be the frozen-background map, $\xi, \xi' \in \mathcal{X}_F$ denote admissible foreground inputs and $\Delta W^{(\mathrm{fg})}$ are foreground-branch updates. Under SMA, semantic leakage is controlled both structurally and quantitatively. In the hard-gating regime, background outputs are conditionally independent of foreground inputs (Eq. 16). In the soft-gating regime with margin $\gamma > 0$, training-time leakage and inference-time sensitivity admit *upper bounds* that decay as $e^{-\gamma}$ (Eq. 17 and Eq. 18).

# F  THEORETICAL PROOFS

We now provide formal proofs for the theoretical results presented in Section E. Our arguments follow the order: Theorem 2, Theorem 1, Corollary 1, and Corollary 2. The main reason for this ordering is that the conditioning analysis of the multi-layer parameterization (Theorem 1) requires control over the cross-domain coupling term $C$, which is precisely provided by the leakage suppression guarantees of SMA in Theorem 2. Corollary 1 then follows as a direct consequence of Theorem 1, while Corollary 2 establishes the empirical risk comparison between multi-layer and single-layer parameterizations. This structure makes explicit the dependency of our optimization guarantees on the semantic isolation ensured by SMA.

## F.1  PROOF OF LEMMA

**Lemma 1.** *Let $p \in \Delta^{d-1}$ and $J = \mathrm{Diag}(p) - pp^\top$. Then $\|J\|_2 \leq \frac{1}{2}$. Moreover, the bound is tight (e.g., for $d = 2$ and $p = (\frac{1}{2}, \frac{1}{2})$).*

*Proof.* $J$ is symmetric and positive semidefinite, since for any $v \in \mathbb{R}^d$,

$$v^\top J v = \sum_{i=1}^{d} p_i v_i^2 - \Big( \sum_{i=1}^{d} p_i v_i \Big)^2 = \mathrm{Var}_p(v) \;\geq\; 0. \tag{19}$$

Thus all eigenvalues of $J$ are real and nonnegative. For $i \neq j$, $J_{ij} = -p_i p_j$ and $J_{ii} = p_i(1 - p_i)$. By the Gershgorin disk theorem (Varga, 2004), every eigenvalue of $J$ lies in

$$\bigcup_{i=1}^{d} \big[\, J_{ii} - R_i,\; J_{ii} + R_i \,\big], \qquad R_i \;=\; \sum_{j \neq i} |J_{ij}| \;=\; p_i(1 - p_i). \tag{20}$$

Hence all eigenvalues lie in

$$\big[\, 0,\; \max_i \{\, J_{ii} + R_i \,\} \,\big] \;=\; \big[\, 0,\; \max_i 2 p_i(1 - p_i) \,\big] \;\subset\; \big[\, 0,\; \tfrac{1}{2} \,\big], \tag{21}$$

since $x(1 - x) \leq \frac{1}{4}$ for all $x \in [0, 1]$. Therefore $\|J\|_2 \leq \frac{1}{2}$. $\qquad\square$

**Lemma 2** (Restricted row-softmax Jacobian). *Let $p \in \Delta^{d-1}$ and $J = \mathrm{Diag}(p) - pp^\top$. For any index subset $F \subseteq \{1, \ldots, d\}$ with mass $s_F := \sum_{j \in F} p_j$, let $P_F$ be the column projector onto $F$. Then*

$$\|J P_F\|_2 \;\leq\; \min\big\{ \tfrac{1}{2},\; 2 s_F \big\}. \tag{22}$$

*Proof.* Take any $u \in \mathbb{R}^d$ supported on $F$ with $\|u\|_2 = 1$. We have

$$Ju \;=\; \mathrm{Diag}(p)\, u - p\,(p^\top u). \tag{23}$$

For the first term,

$$\|\mathrm{Diag}(p)\,u\|_2^2 = \sum_{j \in F} p_j^2 u_j^2 \ \leq \ \Big(\sum_{j \in F} p_j\Big)^2 \sum_{j \in F} u_j^2 = s_F^2, \tag{24}$$

hence $\|\mathrm{Diag}(p)\,u\|_2 \leq s_F$. For the second term,

$$|p^\top u| = \Big|\sum_{j \in F} p_j u_j\Big| \leq \sum_{j \in F} p_j |u_j| \leq s_F \|u\|_\infty \leq s_F \|u\|_2 = s_F, \qquad \|p\|_2 \leq \|p\|_1 = 1. \tag{25}$$

Thus $\|p\,(p^\top u)\|_2 \leq s_F$. By triangle inequality,

$$\|Ju\|_2 \ \leq \ s_F + s_F \ = \ 2s_F. \tag{26}$$

Since $\|J\|_2 \leq \frac{1}{2}$ (see Lemma 1), we obtain

$$\|Ju\|_2 \ \leq \ \min\big\{\tfrac{1}{2},\ 2s_F\big\}. \tag{27}$$

Taking the supremum over unit $u$ supported on $F$ yields the claim. $\qquad\square$

### F.2 Proof of Theorem 2

**Setup.** Let $A_{ij} = \frac{Q_i K_j^\top}{\sqrt{d_k}}$ be the vanilla attention logits between query $i$ and key $j$. Let $z_{ij}$ denote the SMA-biased logits and let

$$\tilde{P}_{ij} := \mathrm{softmax}_j(z_{ij})$$

be the corresponding attention weights (we reserve $A$ for logits and use $\tilde{P}$ for weights to avoid overloading). Write $F$ and $B$ for the sets of foreground and background tokens, respectively. Assume the backbone is deterministic (so functional conditional independences imply conditional mutual information 0) and recall from Lemma 1 that each row-softmax Jacobian satisfies $\|J_i\|_2 \leq \frac{1}{2}$. Define the log-sum-exp margin

$$\gamma \ := \ \inf_{i \in B}\left[\log \sum_{j \in B} e^{z_{ij}} \ - \ \log \sum_{j \in F} e^{z_{ij}}\right], \tag{28}$$

and assume throughout that $\gamma \geq 0$ (SMA bias favors intra-background attention for background queries). Let each linear/convolutional operator have spectral norm $\|W_\ell\|_2$ and each nonlinearity Lipschitz constant $L_\ell$, and define the network Lipschitz constant

$$C_{\mathrm{net}} \ := \ \prod_{\ell=1}^{L} \|W_\ell\|_2\, L_\ell. \tag{29}$$

For a fixed row $i$, let $V_F(i) := [V_j]_{j \in F} \in \mathbb{R}^{d_v \times |F|}$ collect the value vectors on foreground columns and set

$$C_V \ := \ \sup_i \|V_F(i)\|_2 \ < \ \infty, \tag{30}$$

which we absorb into constants below.

**Hard-gating.** If SMA sets all cross-domain logits to $-\infty$, then for any background location $u$ with $\mathrm{dist}(u, F) > R_{\mathrm{eff}}$ the receptive field of $u$ contains no path from any foreground token to $Y(u)$. Hence, under the determinism assumption,

$$I\big(Y(u); X_F \mid X_B\big) \ = \ 0. \tag{31}$$

**Soft-gating.** From the definition of $\gamma$ we have, for any $i \in B$,

$$\sum_{j \in F} e^{z_{ij}} \ \leq \ e^{-\gamma} \sum_{j \in B} e^{z_{ij}}, \tag{32}$$

which yields the bound on normalized foreground attention mass

$$\frac{\sum_{j \in F} e^{z_{ij}}}{\sum_{j \in B \cup F} e^{z_{ij}}} \;\leq\; \frac{e^{-\gamma}}{1 + e^{-\gamma}} \;\leq\; e^{-\gamma}. \tag{33}$$

Moreover, letting $P_F$ denote the column projector onto indices in $F$, Lemma 2 gives the *restricted* Jacobian bound

$$\|J_i P_F\|_2 \;\leq\; \min\left\{\tfrac{1}{2}, \, 2\sum_{j \in F} \tilde{P}_{ij}\right\} \;\leq\; \min\left\{\tfrac{1}{2}, \, \frac{2e^{-\gamma}}{1 + e^{-\gamma}}\right\} \;\leq\; \min\left\{\tfrac{1}{2}, \, 2e^{-\gamma}\right\}. \tag{34}$$

**Training-time leakage bound.** Let the background-branch output be $H_i^{\text{out}} = \sum_j \tilde{P}_{ij} V_j$ and consider the gradient of $\mathcal{L}_{\text{bg}}$ w.r.t. foreground updates $\Delta W^{(\text{fg})}$. By the chain rule and the value-matrix operator norm,

$$\left\|\nabla_{\Delta W^{(\text{fg})}} \mathcal{L}_{\text{bg}}\right\| = \left\|\sum_{j \in F} \frac{\partial \tilde{P}_{ij}}{\partial \Delta W^{(\text{fg})}} V_j\right\| \;=\; \left\|V_F(i) \frac{\partial \tilde{P}_{i,F}}{\partial \Delta W^{(\text{fg})}}\right\|_2 \tag{35}$$

$$\leq \|V_F(i)\|_2 \cdot \left\|\frac{\partial \tilde{P}_i}{\partial \Delta W^{(\text{fg})}} P_F\right\|_2 \;=\; C_V \cdot \left\|J_i \frac{\partial z_i}{\partial \Delta W^{(\text{fg})}} P_F\right\|_2 \tag{36}$$

$$\leq C_V \cdot \|J_i P_F\|_2 \cdot \left\|\frac{\partial z_i}{\partial \Delta W^{(\text{fg})}}\right\|_2 \tag{37}$$

$$\leq \min\left\{\tfrac{1}{2}, \, \frac{2e^{-\gamma}}{1 + e^{-\gamma}}\right\} C_{\text{net}} \, C_V. \tag{38}$$

**Inference-time sensitivity.** Let $g : \mathcal{X}_F \to \mathcal{Y}_B$ be the background predictor with background fixed. For any $\xi, \xi' \in \mathcal{X}_F$,

$$\|g(\xi) - g(\xi')\| = \left\|\int_0^1 J_g\big(\xi' + s(\xi - \xi')\big)\, (\xi - \xi') \, ds\right\| \tag{39}$$

$$\leq \left(\sup_\zeta \|J_g(\zeta)\|_2\right) \|\xi - \xi'\| \tag{40}$$

$$\leq \min\left\{\tfrac{1}{2}, \, \frac{2e^{-\gamma}}{1 + e^{-\gamma}}\right\} C_{\text{net}} \, C_V \|\xi - \xi'\|. \tag{41}$$

Here the last step mirrors the training-time decomposition: the only pathway for foreground perturbations to affect background outputs is via cross-domain attention, whose *restricted* Jacobian spectral norm contributes the $\min\{\tfrac{1}{2}, \frac{2e^{-\gamma}}{1+e^{-\gamma}}\}$ factor, and whose remaining propagation is absorbed into $C_{\text{net}} C_V$.

Combining the hard-gating independence with the exponentially suppressed coupling bounds equation 38 and equation 41 completes the proof.

### F.3 PROOF OF THEOREM 1

**Setup.** Assume each backbone operator $W_\ell$ has finite spectral norm $\|W_\ell\|_2$ and each nonlinearity $\sigma_\ell$ is $L_\ell$-Lipschitz. Then the overall network is Lipschitz with constant

$$C_{\text{net}} \;:=\; \prod_{\ell=1}^{L} \|W_\ell\|_2 \, L_\ell. \tag{42}$$

We also assume bounded background-loss derivatives in output space:

$$\|\nabla_{Y_{\text{bg}}} \mathcal{L}_{\text{bg}}\| \;\leq\; C_{\text{grad}}, \qquad \|\nabla_{Y_{\text{bg}}}^2 \mathcal{L}_{\text{bg}}\|_2 \;\leq\; C_{\text{loss}}. \tag{43}$$

The total training loss reads

$$\mathcal{L}_{\text{total}}(\theta_{\text{fg}}, \theta_{\text{bg}}) \;=\; \lambda_s \sum_{k=1}^{N} \mathcal{L}_k(\theta_{\text{fg}}) \;+\; \mathcal{L}_{\text{bg}}(\theta_{\text{fg}}, \theta_{\text{bg}}), \tag{44}$$

where the dependence of $\mathcal{L}_{\mathrm{bg}}$ on $\theta_{\mathrm{fg}}$ captures possible shared computations (e.g., attention-mediated mixing). Let the block Hessian be

$$H \;=\; \nabla^2_{(\theta_{\mathrm{fg}}, \theta_{\mathrm{bg}})} \mathcal{L}_{\mathrm{total}} \;=\; \begin{bmatrix} H_{\mathrm{fg}} & C \\ C^\top & H_{\mathrm{bg}} \end{bmatrix}, \qquad C \;=\; \frac{\partial}{\partial \theta_{\mathrm{bg}}} \Big( \nabla_{\theta_{\mathrm{fg}}} \mathcal{L}_{\mathrm{total}} \Big). \tag{45}$$

Let $\gamma$ be the cross-domain margin defined in Eq. 8, and *assume* $\gamma \geq 0$ (SMA bias favors intra-background attention for background queries).

**Hard-gating.** If the foreground and background branches use disjoint parameters and there is no computational path from $\theta_{\mathrm{fg}}$ to $\mathcal{L}_{\mathrm{bg}}$ (and vice versa), then $\mathcal{L}_{\mathrm{bg}}(\theta_{\mathrm{fg}}, \theta_{\mathrm{bg}})$ reduces to $\mathcal{L}_{\mathrm{bg}}(\theta_{\mathrm{bg}})$ and

$$\frac{\partial}{\partial \theta_{\mathrm{bg}}} \nabla_{\theta_{\mathrm{fg}}} \mathcal{L}_{\mathrm{total}} \;=\; 0, \qquad \frac{\partial}{\partial \theta_{\mathrm{fg}}} \nabla_{\theta_{\mathrm{bg}}} \mathcal{L}_{\mathrm{total}} \;=\; 0, \tag{46}$$

hence

$$C \;=\; 0, \qquad H \;=\; \begin{bmatrix} H_{\mathrm{fg}} & 0 \\ 0 & H_{\mathrm{bg}} \end{bmatrix}. \tag{47}$$

Therefore the parameter updates are decoupled:

$$\theta_{\mathrm{fg}}^{(t+1)} \;=\; \theta_{\mathrm{fg}}^{(t)} - \eta \, \lambda_s \sum_{k=1}^{N} \nabla_{\theta_{\mathrm{fg}}} \mathcal{L}_k(\theta_{\mathrm{fg}}^{(t)}), \qquad \theta_{\mathrm{bg}}^{(t+1)} \;=\; \theta_{\mathrm{bg}}^{(t)} - \eta \, \nabla_{\theta_{\mathrm{bg}}} \mathcal{L}_{\mathrm{bg}}(\theta_{\mathrm{bg}}^{(t)}). \tag{48}$$

**Soft-gating.** Suppose there exist shared computations causing $\mathcal{L}_{\mathrm{bg}}$ to depend on $\theta_{\mathrm{fg}}$ through attention-mediated features (e.g., shared $Q/K/V$ projections, shared normalization layers, or residual mixing). Let $Y_{\mathrm{bg}}$ denote the background-branch output; by the chain rule,

$$\begin{aligned} C &= \frac{\partial}{\partial \theta_{\mathrm{bg}}} \Big( \nabla_{\theta_{\mathrm{fg}}} \mathcal{L}_{\mathrm{bg}} \Big) = \frac{\partial}{\partial \theta_{\mathrm{bg}}} \Big( J^\top_{Y_{\mathrm{bg}}, \theta_{\mathrm{fg}}} \nabla_{Y_{\mathrm{bg}}} \mathcal{L}_{\mathrm{bg}} \Big) \\ &= \underbrace{\frac{\partial J^\top_{Y_{\mathrm{bg}}, \theta_{\mathrm{fg}}}}{\partial \theta_{\mathrm{bg}}} \nabla_{Y_{\mathrm{bg}}} \mathcal{L}_{\mathrm{bg}}}_{\text{path via feature Jacobian}} + \underbrace{J^\top_{Y_{\mathrm{bg}}, \theta_{\mathrm{fg}}} \nabla^2_{Y_{\mathrm{bg}}} \mathcal{L}_{\mathrm{bg}} \, J_{Y_{\mathrm{bg}}, \theta_{\mathrm{bg}}}}_{\text{path via loss curvature}}. \end{aligned} \tag{49}$$

By Theorem 2 and the assumption $\gamma \geq 0$, the normalized attention mass on foreground keys satisfies

$$\sum_{j \in F} \tilde{P}_{ij} \;\leq\; e^{-\gamma}. \tag{50}$$

Consequently, foreground-mediated Jacobians entering the background path are exponentially suppressed. Moreover, letting $P_F$ denote the column projector onto indices in $F$, Lemma 2 together with equation 50 implies that the restricted row-softmax Jacobian satisfies $\|J_i P_F\|_2 \leq \min\{\frac{1}{2}, 2e^{-\gamma}/(1 + e^{-\gamma})\} \leq 2e^{-\gamma}$. Propagating through the remaining operators, we obtain

$$\|J_{Y_{\mathrm{bg}}, \theta_{\mathrm{fg}}}\|_2 \;\leq\; C^{(1)}_{\mathrm{net}} \, e^{-\gamma}, \qquad \left\| \frac{\partial J_{Y_{\mathrm{bg}}, \theta_{\mathrm{fg}}}}{\partial \theta_{\mathrm{bg}}} \right\|_2 \;\leq\; C^{(2)}_{\mathrm{net}} \, e^{-\gamma}, \tag{51}$$

where $C^{(1)}_{\mathrm{net}}, C^{(2)}_{\mathrm{net}}$ depend only on layer operator norms, nonlinearity Lipschitz constants, and the fact that softmax has globally bounded first- and second-order derivatives (these universal constants are absorbed into $C^{(1)}_{\mathrm{net}}, C^{(2)}_{\mathrm{net}}$). Moreover,

$$\|J_{Y_{\mathrm{bg}}, \theta_{\mathrm{bg}}}\|_2 \;\leq\; C_{\mathrm{net}}. \tag{52}$$

Combining equation 49–equation 52 with equation 43 yields

$$\|C\|_2 \leq C^{(2)}_{\mathrm{net}} e^{-\gamma} \, C_{\mathrm{grad}} \;+\; (C^{(1)}_{\mathrm{net}} e^{-\gamma}) \, C_{\mathrm{loss}} \, (C_{\mathrm{net}}) \;\leq\; \Big( C^{(2)}_{\mathrm{net}} C_{\mathrm{grad}} + C^{(1)}_{\mathrm{net}} C_{\mathrm{loss}} C_{\mathrm{net}} \Big) e^{-\gamma} \;=:\; K \, e^{-\gamma}. \tag{53}$$

Therefore,

$$\|C\|_2 \;\leq\; K \, e^{-\gamma}, \tag{54}$$

where $K$ depends only on operator norms of the backbone layers, Lipschitz constants of nonlinearities, the softmax Jacobian bound, and the background-loss smoothness/gradient bounds $(C_{\mathrm{loss}}, C_{\mathrm{grad}})$. This completes the proof.

### F.4   PROOF OF COROLLARY 1

Let

$$H = \begin{pmatrix} H_{\mathrm{fg}} & C \\ C^\top & H_{\mathrm{bg}} \end{pmatrix}, \tag{55}$$

with fixed diagonal blocks $H_{\mathrm{fg}}, H_{\mathrm{bg}}$ that are SPD, and define

$$m := \min\{\lambda_{\min}(H_{\mathrm{fg}}), \lambda_{\min}(H_{\mathrm{bg}})\} > 0, \qquad M := \max\{\lambda_{\max}(H_{\mathrm{fg}}), \lambda_{\max}(H_{\mathrm{bg}})\}, \tag{56}$$

and $H_0 := \mathrm{diag}(H_{\mathrm{fg}}, H_{\mathrm{bg}})$. Write

$$E := \begin{pmatrix} 0 & C \\ C^\top & 0 \end{pmatrix}. \tag{57}$$

By Weyl's perturbation inequality (spectral-norm form) for symmetric matrices,

$$\lambda_{\max}(H_0 + E) \le \lambda_{\max}(H_0) + \|E\|_2 = M + \|C\|_2, \tag{58}$$
$$\lambda_{\min}(H_0 + E) \ge \lambda_{\min}(H_0) - \|E\|_2 = m - \|C\|_2, \tag{59}$$

where we used $\|E\|_2 = \|C\|_2$. Hence, whenever $\|C\|_2 < m$,

$$\kappa(H) = \frac{\lambda_{\max}(H)}{\lambda_{\min}(H)} \le \frac{M + \|C\|_2}{m - \|C\|_2}. \tag{60}$$

If SMA enforces $\|C\|_2 \le K\,e^{-\gamma}$ for some $K > 0$, then for $Ke^{-\gamma} < m$,

$$\kappa(H) \le \frac{M + Ke^{-\gamma}}{m - Ke^{-\gamma}} = \kappa(H_0)\left(1 + O(e^{-\gamma})\right), \tag{61}$$

so this Weyl-type conditioning upper bound decays exponentially in $\gamma$ and converges to $\kappa(H_0) = M/m$ as $\gamma \to \infty$.

Moreover, for any coupled baseline $H_{\mathrm{single}}$ with the same diagonal blocks and coupling $\widehat{C}$ satisfying $\|\widehat{C}\|_2 \ge \|C\|_2$, monotonicity of $f(x) = (M + x)/(m - x)$ on $(-m, m)$ gives

$$\kappa(H) \le \frac{M + \|C\|_2}{m - \|C\|_2} \le \frac{M + \|\widehat{C}\|_2}{m - \|\widehat{C}\|_2}. \tag{62}$$

### F.5   PROOF OF COROLLARY 2

Consider a linearized quadratic approximation with a matched $\ell_2$ regularization budget. Let $\Phi_{F_k}$ and $\Phi_B$ denote the fixed design matrices under the linearization for the $k$-th foreground and the background, respectively, with targets $y_{F_k}, y_B$. The single-layer empirical risk is

$$J_{\mathrm{single}}(\theta) = \sum_{k=1}^{N} \|\Phi_{F_k}\theta - y_{F_k}\|_2^2 + \|\Phi_B\theta - y_B\|_2^2 + \lambda\|\theta\|_2^2. \tag{63}$$

For the multi-layer parameterization, distribute the same total budget $\lambda$ across the background and the $N$ foreground blocks via nonnegative weights $\lambda_B, \lambda_1, \ldots, \lambda_N$ satisfying

$$\lambda_B + \sum_{k=1}^{N} \lambda_k = \lambda, \tag{64}$$

and define

$$J_{\mathrm{multi}}(w_B, \{w_k\}) = \sum_{k=1}^{N} \|\Phi_{F_k}w_k - y_{F_k}\|_2^2 + \|\Phi_B w_B - y_B\|_2^2 + \lambda_B\|w_B\|_2^2 + \sum_{k=1}^{N} \lambda_k\|w_k\|_2^2. \tag{65}$$

(In particular, the uniform split $\lambda_B = \lambda_k = \lambda/(N+1)$ also suffices.)

If we constrain the multi-layer parameters by setting $w_B = w_1 = \cdots = w_N = \theta$, then by the budget-matching identity,

$$J_{\text{multi}}(\theta, \ldots, \theta) = \sum_{k=1}^{N} \|\Phi_{F_k}\theta - y_{F_k}\|_2^2 + \|\Phi_B\theta - y_B\|_2^2 + \left(\lambda_B + \sum_{k=1}^{N} \lambda_k\right)\|\theta\|_2^2$$

$$= \sum_{k=1}^{N} \|\Phi_{F_k}\theta - y_{F_k}\|_2^2 + \|\Phi_B\theta - y_B\|_2^2 + \lambda\|\theta\|_2^2$$

$$= J_{\text{single}}(\theta). \tag{66}$$

Therefore the single-layer case is recovered as a special instance of the multi-layer formulation, and taking minima yields

$$\min_{w_B, \{w_k\}} J_{\text{multi}}(w_B, \{w_k\}) \leq \min_{\theta} J_{\text{single}}(\theta). \tag{67}$$

Hence, under a matched $\ell_2$ budget, the optimal empirical risk of the multi-layer parameterization is no larger than that of the single-layer baseline, which establishes Corollary 2.

### F.6 PROOFS FOR SEC. 5.1

We work row-wise. For a row with logits $a \in \mathbb{R}^d$ and softmax $p = \text{softmax}(a)$, let $m \in \{0,1\}^d$ indicate cross-domain (foreground) columns $F = \{j : m_j = 1\}$ and $B$ be the complement. A logit bias $-\beta m$ ($\beta \geq 0$) gives $\tilde{a} = a - \beta m$ and $\tilde{p} = \text{softmax}(\tilde{a})$. Denote the row-wise cross-domain projector by $\mathcal{T}$, i.e., $(\mathcal{T}(p^\top))_j = p_j$ if $j \in F$ and 0 otherwise. For a full attention matrix $P$ (row-stacked $p^\top$) with mask $M$, write $\mathcal{T}(P) = P \odot M$. We first control the cross-domain *mass* after adding the bias. Writing $Z_F = \sum_{j \in F} e^{a_j}$ and $Z_B = \sum_{j \in B} e^{a_j}$, the bias gives $Z'_F = e^{-\beta}Z_F$ and $Z'_B = Z_B$, hence

$$\sum_{j \in F} \tilde{p}_j = \frac{e^{-\beta}Z_F}{Z_B + e^{-\beta}Z_F} \leq e^{-\beta}\frac{Z_F}{Z_B + Z_F} = e^{-\beta}\sum_{j \in F} p_j.$$

Thus, for each row, the cross-domain mass (the $\ell_1$-norm on $F$) contracts by at least $e^{-\beta}$, and since $\mathcal{T}(P)$ is entrywise nonnegative,

$$\left\|\mathcal{T}(\tilde{P})\right\|_\infty = \max_i \sum_j \left(\mathcal{T}(\tilde{P})\right)_{ij} \leq e^{-\beta}\max_i \sum_j \left(\mathcal{T}(P)\right)_{ij} = e^{-\beta}\left\|\mathcal{T}(P)\right\|_\infty. \tag{68}$$

This is a clean multiplicative contraction in the induced $\infty$-norm (row-sum norm). In parallel, a Jacobian-based estimate provides a complementary *additive* control in $\ell_2$: with $J_{\text{sm}}(a) = \text{Diag}(p) - pp^\top$ the row-softmax Jacobian and $\|J_{\text{sm}}(a)\|_2 \leq \frac{1}{2}$ (Lemma 1), along the path $a^{(s)} = a - s\beta m$ we have

$$\tilde{p} - p = \int_0^1 J_{\text{sm}}(a^{(s)})(-\beta m)\, ds, \tag{69}$$

$$\Rightarrow \left\|\mathcal{T}(\tilde{p}^\top) - \mathcal{T}(p^\top)\right\|_2 \leq \int_0^1 \left\|J_{\text{sm}}(a^{(s)})\right\|_2 \beta \|m\|_2\, ds \leq \tfrac{1}{2}\beta\|m\|_2, \tag{70}$$

which is useful for small $\beta$ but does not yield a uniform multiplicative $1/2$-contraction. Finally, for a composition of $L$ attention blocks across $T$ diffusion steps with biases $\{\beta_{\ell,t}\}$, applying equation 68 to each block and using submultiplicativity of induced norms gives

$$\left\|\mathcal{T}_{\text{MILD}}\right\|_\infty \leq \prod_{\ell=1}^{L}\prod_{t=1}^{T} e^{-\beta_{\ell,t}}\left\|\mathcal{T}(P_{\text{base}})\right\|_\infty$$

$$= \exp\left(-\sum_{\ell=1}^{L}\sum_{t=1}^{T}\beta_{\ell,t}\right)\left\|\mathcal{T}(P_{\text{base}})\right\|_\infty$$

$$= e^{-\bar{\beta}LT}\left\|\mathcal{T}(P_{\text{base}})\right\|_\infty, \tag{71}$$

where $\bar{\beta} = \frac{1}{LT}\sum_{\ell,t}\beta_{\ell,t}$ and $P_{\text{base}}$ is the end-to-end attention operator of the same stack *without* SMA biases.

## G  MORE EXPERIMENTS

### G.1  MORE EXPERIMENTAL SETUP

**Implementation Details**  We train MILD LoRA on 20,000 images, including 10,000 real-world pairs from our curated MILD dataset and 10,000 synthetic samples generated by compositing segmented humans onto OpenImages backgrounds with random cropping and mask augmentation. The baseline model (Podell et al., 2023) is optimized using AdamW (Loshchilov & Hutter, 2019) with a learning rate of $1e{-}6$, batch size 4, and trained for 20,000 iterations. Human Morphology Guidance leverages pose maps from OpenPose (Cao et al., 2017) and parsing maps from SCHP (Li et al., 2020b), preprocessed prior to training. A staged training scheme is adopted: the background branch is trained first, followed by a linear ramp-up of foreground losses after 8,000 steps. The model employs a UNet backbone within a latent diffusion framework, equipped with rank-16 LoRA modules (scaling factor 16). Inference uses 25 DDIM steps at $512 \times 512$ resolution. All experiments are conducted on NVIDIA $A$800 GPUs.

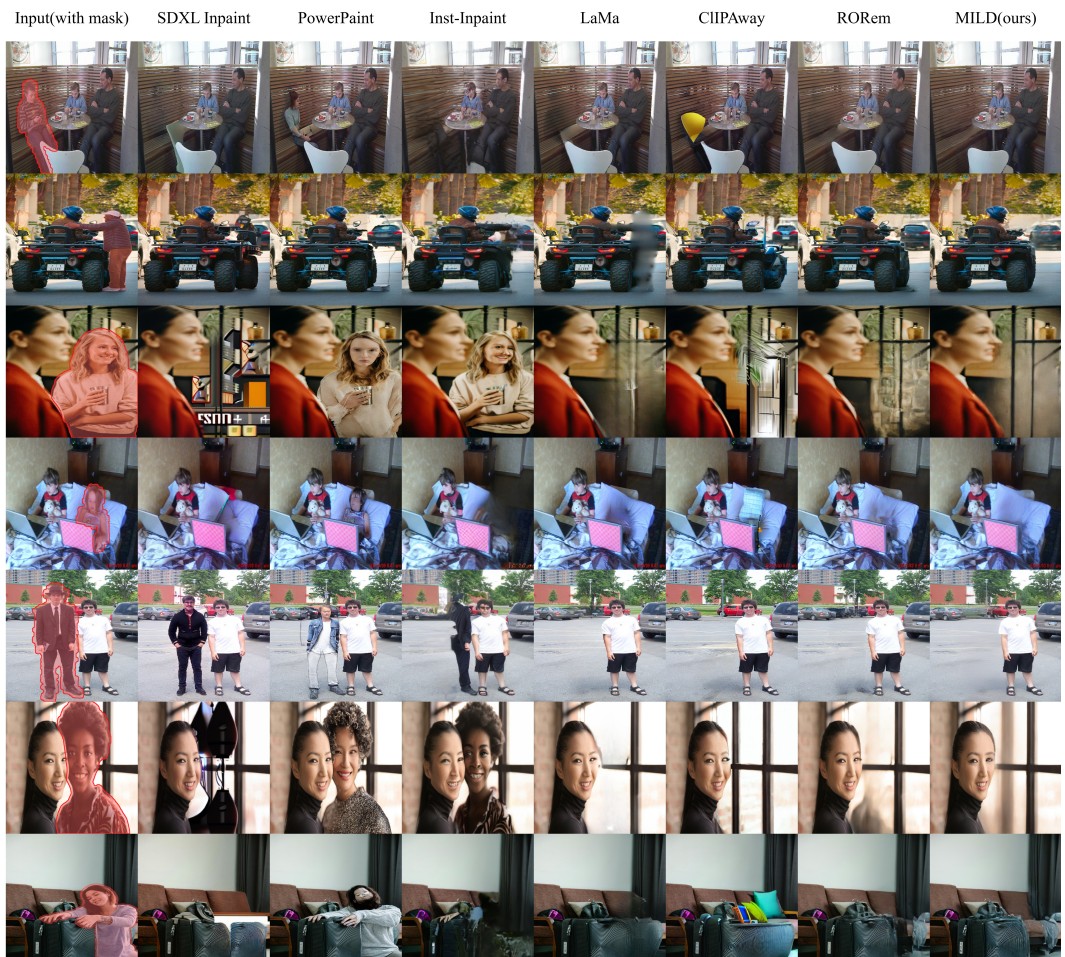

Figure 10:  More qualitative comparisons between MILD and other methods on MILD dataset.

### G.2  MORE QUANTITATIVE RESULTS

To further validate the robustness and generalizability of our method, we introduce the Structural Similarity Index Measure (SSIM) as an additional evaluation metric. Unlike FID, LPIPS, or CLIP-

| Method | FID↓ | LPIPS↓ | DINO↑ | CLIP↑ | PSNR↑ | SSIM↑ |
|---|---|---|---|---|---|---|
| SDXL Inpaint | 53.45 | 0.242 | 0.8398 | 0.8331 | 18.22 | 0.6984 |
| PowerPaint | 48.04 | 0.207 | 0.8901 | 0.8668 | 20.27 | 0.7569 |
| LaMa | 35.02 | 0.164 | 0.8919 | 0.8872 | 24.13 | 0.7984 |
| CLIPAway | 48.86 | 0.230 | 0.8647 | 0.8605 | 19.49 | 0.7046 |
| RoRem | 40.41 | 0.196 | 0.8888 | 0.8810 | 22.35 | 0.7521 |
| **MILD (Ours)** | **24.20** | **0.093** | **0.9703** | **0.9499** | **26.24** | **0.8341** |

Table 3: Quantitative comparison of MILD and other methods on MILD dataset. Best results are **bold**, second best are underlined.

| Method | FID↓ | LPIPS↓ | DINO↑ | CLIP↑ | PSNR↑ | SSIM↑ |
|---|---|---|---|---|---|---|
| SDXL Inpaint | 26.22 | 0.1346 | 0.8741 | 0.8751 | 21.88 | 0.8290 |
| PowerPaint | 10.56 | 0.0488 | 0.9649 | 0.9678 | 31.04 | **0.9638** |
| Inst-Inpaint | 11.42 | 0.4100 | 0.7400 | 0.8207 | 23.75 | 0.8023 |
| LaMa | **10.38** | **0.0470** | 0.9192 | 0.9293 | 31.83 | 0.9461 |
| CLIPAway | 22.73 | 0.1283 | 0.8973 | 0.8985 | 23.59 | 0.8276 |
| RoRem | 18.23 | 0.0982 | 0.9095 | 0.9148 | 26.59 | 0.8713 |
| **MILD (Ours)** | 17.86 | 0.0668 | **0.9829** | **0.9700** | **32.14** | 0.9515 |

Table 4: Quantitative comparison of object removal methods on the **OpenImages** dataset. Best results are **bold**, second best are underlined.

based perceptual scores, SSIM captures low-level structural consistency between the generated and ground-truth images, providing complementary insight into image fidelity.

As summarized in Table 3 (MILD dataset), Table 4 (OpenImages dataset) and Table 5 (COCO Lin et al. (2014) and ADE20K Zhou et al. (2017) dataset), our method consistently outperforms prior approaches across all major metrics, including the newly introduced SSIM. This further confirms that MILD not only produces perceptually realistic outputs but also preserves structural coherence with the surrounding context.

The inclusion of SSIM highlights MILD's ability to maintain both global semantics and local structural details, reinforcing its effectiveness in complex object removal scenarios.

## G.3    MORE QUALITATIVE RESULTS

To further demonstrate the effectiveness of our approach, we present additional qualitative comparisons against state-of-the-art baselines in diverse and challenging scenarios. These include scenes with multiple interacting persons, complex occlusions, cluttered backgrounds, and fine structural details such as limbs, shadows, and object boundaries.

As shown in Figure 10, our method produces more semantically consistent and visually coherent results compared to existing methods. In particular, our model excels at generating plausible textures, maintaining global scene structure, and avoiding common failure cases such as blurry artifacts, object duplication, or semantic distortions. These results further validate the benefits of our layered generation strategy, human-centric guidance, and attention modulation.

Furthermore, Figure 11 and Figure 12 illustrate MILD's qualitative results on COCO and ADE20K (animals, furniture, etc.), showing promising generalization to non-human scenes.

We also demonstrate the pixel-level difference betweeen the original inputs and our edited results. As shown in figure 13, the difference maps(col 3) and their enhanced visualizations(col 4) highlight that our method modifies the target object regions while preserving the surrounding background. This indicates the effectiveness of our approach in achieving precise localized editing, with only minor adjustments to the background.

| Method | COCO | | | | | ADE20K | | | | |
| --- | --- | --- | --- | --- | --- | --- | --- | --- | --- | --- |
| | FID↓ | LPIPS↓ | DINO↑ | CLIP↑ | PSNR↑ | FID↓ | LPIPS↓ | DINO↑ | CLIP↑ | PSNR↑ |
| SDXL Inpaint | 33.62 | 0.0829 | 0.8572 | 0.9135 | 13.78 | 27.22 | 0.0697 | 0.8896 | 0.9110 | 13.68 |
| PowerPaint | 88.82 | 0.0681 | 0.8997 | 0.9557 | 9.75 | 72.61 | 0.0423 | **0.9386** | 0.9691 | 10.78 |
| LaMa | 102.71 | 0.0667 | 0.8749 | 0.9493 | 11.07 | 66.59 | 0.0414 | 0.9272 | 0.9686 | 11.41 |
| CLIPAway | 35.49 | 0.0750 | 0.8598 | 0.9176 | 14.85 | 29.26 | 0.0651 | 0.8877 | 0.9128 | 14.34 |
| RoRem | **27.35** | **0.0487** | 0.8865 | 0.9298 | **20.32** | **25.52** | 0.0485 | 0.9052 | 0.9222 | 18.51 |
| **MILD (Ours)** | 32.13 | 0.0572 | **0.9248** | **0.9695** | 19.83 | 29.16 | **0.0408** | 0.9276 | **0.9704** | **19.48** |

Table 5: Quantitative comparison of object removal methods on the **COCO** dataset and **ADE20K** dataset. Best results are **bold**, second best are underlined.

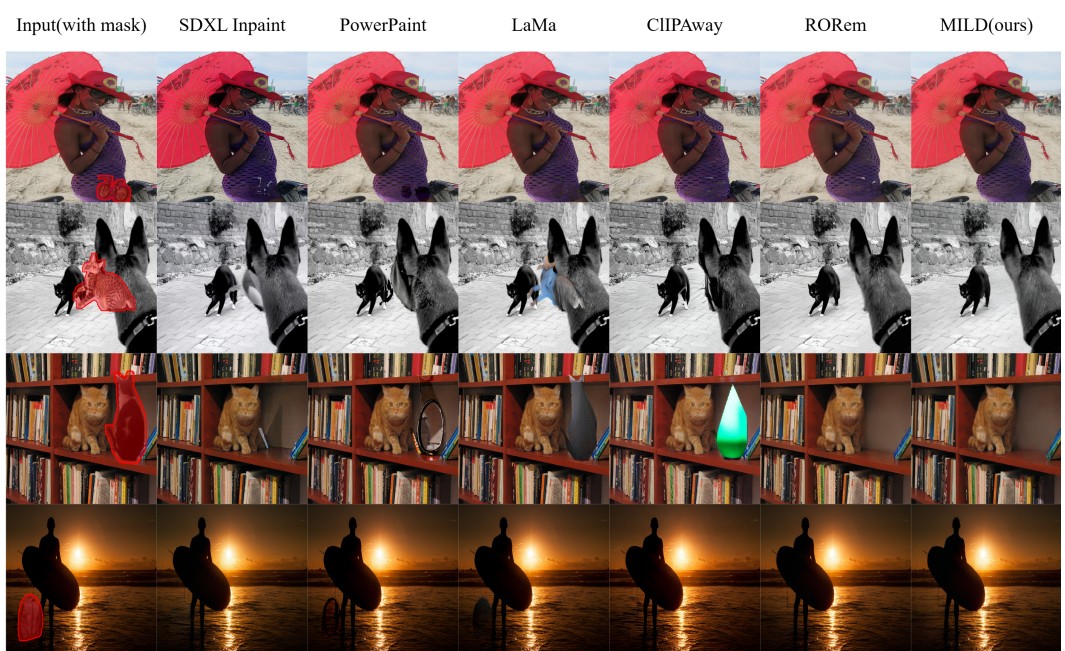

Figure 11: More qualitative comparisons between MILD and other methods on COCO dataset.

### G.4 MORE ABLATION STUDY

As shown in Table G.4, switching from the vanilla diffusion backbone (row 1) to our multi-layer backbone (row 2) already yields a large gain, so all subsequent variants are built upon this backbone. Adding the SMA module on top of it further improves global consistency (row 5 vs. row 2).

To better understand the role of each morphological prior in HMG, we then conduct ablations that isolate the pose-only and parsing-only configurations, with results summarized in rows 3–4. Removing both priors leads to a substantial performance drop across all metrics (e.g., FID +29.25 and PSNR –8.02 relative to the full model), showing that human removal is hard without structured guidance. Enabling only pose or only parsing brings moderate gains but still lags behind full HMG. The pose-only variant mainly helps global structure, reflected in stronger DINO alignment (+0.0962 over the no-prior baseline), but shows higher LPIPS and weaker CLIP scores, indicating less accurate boundaries. The parsing-only variant gives better boundary-level metrics (LPIPS 0.133 vs. 0.142 for pose-only) but is less robust under complex body configurations, as seen from its higher FID and lower DINO/CLIP. Combining both priors in full HMG achieves the best results on all metrics, suggesting that pose stabilizes global geometry while parsing provides fine-grained spatial constraints.

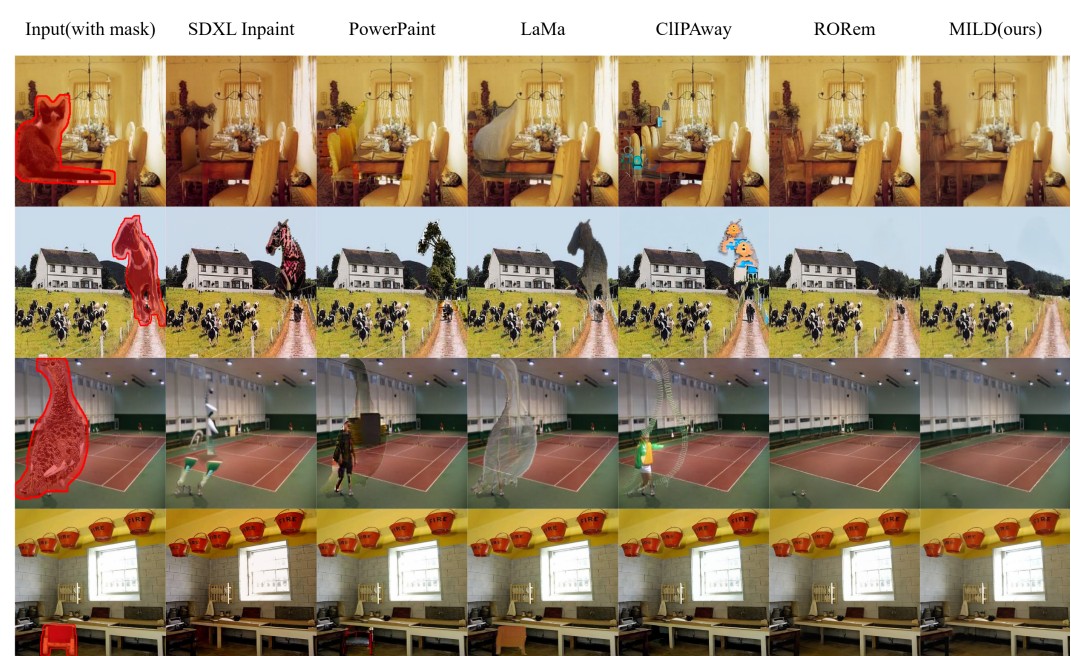

Input(with mask) SDXL Inpaint PowerPaint LaMa ClIPAway RORem MILD(ours)

Figure 12: More qualitative comparisons between MILD and other methods on ADE20K dataset.

| Backbone | HMG | SMA | Full | FID↓ | Δ | LPIPS↓ | Δ | DINO↑ | Δ | CLIP↑ | Δ | PSNR↑ | Δ |
|---|---|---|---|---|---|---|---|---|---|---|---|---|---|
| ✗ | ✗ | ✗ | ✗ | 53.45 | +29.25 | 0.242 | +0.149 | 0.8398 | -0.1305 | 0.8331 | -0.1168 | 18.22 | -8.02 |
| ✔ | ✗ | ✗ | ✗ | 29.02 | +4.82 | 0.142 | +0.049 | 0.9110 | -0.0593 | 0.9025 | -0.0474 | 23.45 | -2.79 |
| ✔ | ✔(without parsing) | ✗ | ✗ | 27.48 | +3.28 | 0.131 | +0.038 | 0.9360 | -0.0343 | 0.9287 | +0.0212 | 24.26 | -1.98 |
| ✔ | ✔(without pose) | ✗ | ✗ | 26.79 | +2.59 | 0.133 | +0.040 | 0.9327 | -0.0376 | 0.9221 | +0.0278 | 24.13 | -2.11 |
| ✔ | ✔ | ✗ | ✗ | 26.10 | +1.90 | 0.120 | +0.027 | 0.9440 | -0.0263 | 0.9312 | -0.0187 | 24.67 | -1.57 |
| ✔ | ✗ | ✔ | ✗ | 25.10 | +0.90 | 0.105 | +0.012 | 0.9380 | -0.0323 | 0.9227 | -0.0272 | 25.21 | -1.03 |
| ✔ | ✔ | ✔ | ✔ | **24.20** | – | **0.093** | – | **0.9703** | – | **0.9499** | – | **26.24** | – |

Table 6: Ablation study on the MILD dataset comparing the SDXL baseline, the model with the MILD backbone, and incremental integration of HMG variants and SMA.

### G.5 INFERENCE EFFICIENCY.

Table G.5 summarizes the average inference time per image at $512 \times 512$ resolution using an A800 GPU with 25 diffusion steps. Despite incorporating a layered, multi-branch generation strategy, **MILD** maintains efficient runtime performance. Due to the shared UNet backbone and lightweight LoRA-based adaptation, our method achieves instance-aware generation without redundant computation overhead.

Compared to other diffusion-based baselines such as PowerPaint, CLIPAway, and RoRem, MILD demonstrates notably faster inference. While it supports text prompts to enhance generation precision, the impact on speed is minimal, resulting in comparable runtime to the SDXL Inpainting baseline. Although LaMa achieves the fastest runtime overall owing to its CNN-based architecture, it compromises significantly on visual fidelity and semantic accuracy, as evidenced by its performance in Table 4. Moreover, although MILD incurs a marginally higher peak memory cost than competing methods, the overall requirement remains well within a practical range (below 10 GB). Overall, MILD offers a favorable trade-off between generation quality, runtime and memory efficiency.

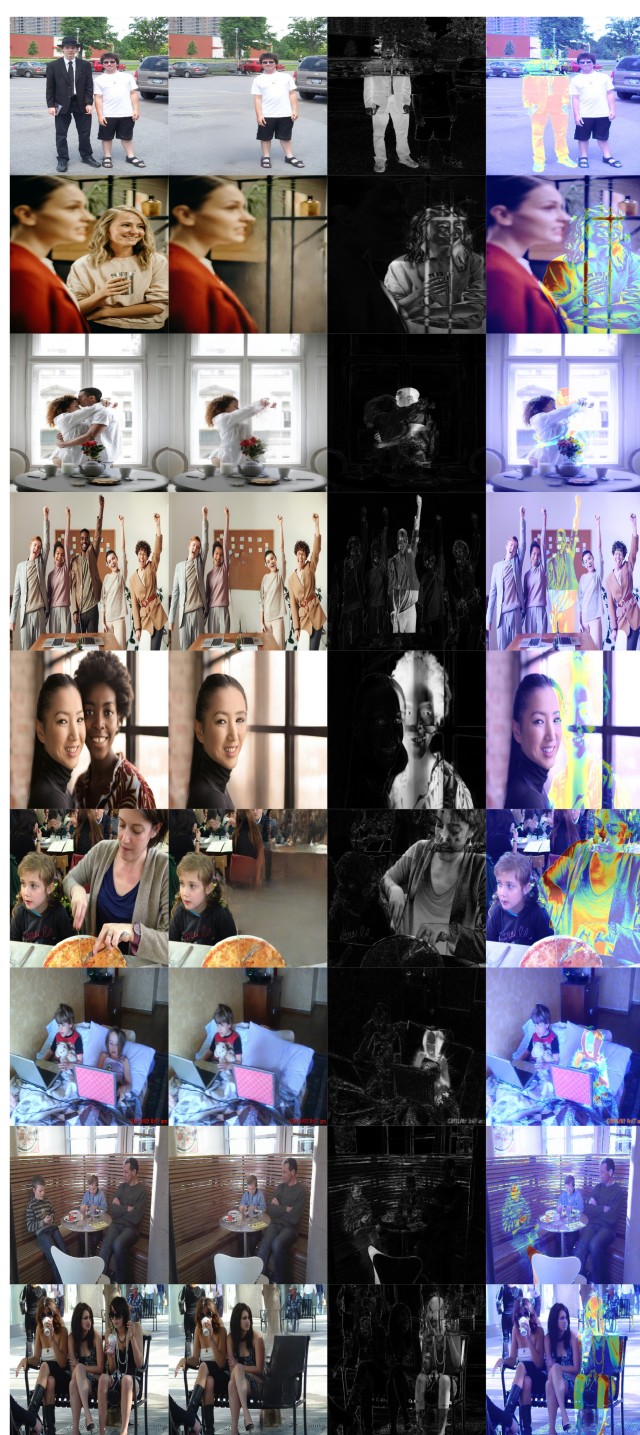

Figure 13: More qualitative results between before and after mild editing.

### G.6 AI EVALUATION.

**Implementation details** We evaluate perceptual quality using both AI-based and human-based protocols. For the AI evaluation, we employ GPT-4o to assess 100 randomly selected image pairs. Given a standardized prompt, GPT-4o performs a side-by-side visual inspection of the original and edited images and assigns four aspect-wise scores including *naturalness*, *semantic consistency*, *vi-*

| Method | Guidance Type | Time (s) | Memory (GB) |
|---|---|---|---|
| SDXL Inpaint | Text + Mask | 1.79 | 7.64 |
| PowerPaint | Text + Mask | 6.21 | **2.02** |
| Inst-Inpaint | Text-only | 2.21 | – |
| LaMa | Mask-only | **1.50** | 5.89 |
| CLIPAway | Text + Mask | 4.76 | 7.65 |
| RoRem | Text + Mask | 4.15 | 7.64 |
| **MILD (Ours)** | Text + Mask | 1.91 | 9.36 |

Table 7: Average inference time per image and peak GPU memory usage.

*sual harmony*, and *artifact suppression*, each on a 1–10 scale. These are then averaged to produce an overall perceptual score. The model also provides a binary decision indicating whether the object removal is **successful** or **unsuccessful**.

For the human evaluation, we recruit 20 participants to rate the same 100 image pairs. Each participant is asked to consider the above four aspects holistically and assign a single overall quality score between 1 and 10. Additionally, they indicate whether the result is **successful** or **unsuccessful**, based on the overall plausibility and absence of noticeable artifacts. Final human scores are computed by averaging all participant ratings, and success rate is calculated as the percentage of participants marking each result as successful.

**AI & Human Evaluation Protocol**   To assess the perceptual quality of human erasing, we employ a unified evaluation protocol involving both AI-based and human-based assessments. In both cases, evaluators are presented with side-by-side comparisons of the original image (with the human present) and the edited result, and are asked to judge the visual plausibility of the removal without access to ground truth.

For the AI evaluation, we use GPT-4o to score each image across four criteria, adapted from standard visual editing assessments: *naturalness*, *semantic consistency*, *visual harmony*, and *artifact suppression*. Each aspect is rated on a 1–10 scale, and the average of these scores is reported as the overall perceptual score. Additionally, GPT-4o provides a binary decision indicating whether the removal is considered **successful** or **unsuccessful**, based on visual realism and contextual coherence.

- **Naturalness:** Whether the filled region appears visually plausible and free from obvious artifacts.
- **Semantic Consistency:** Whether the inpainted content aligns semantically with the surrounding scene.
- **Visual Harmony:** Whether lighting, textures, and shadows in the edited region are consistent with the rest of the image.
- **Artifact Suppression:** Whether the result avoids seams, blurs, distortions, or other visual artifacts.

In the human evaluation, we recruit 20 participants to assess the same set of 100 images. Unlike the AI annotator, human raters do not score each aspect individually. Each image receives multiple independent ratings to reduce individual preference bias. Instead, they are instructed to consider all four criteria jointly and provide a single overall quality score (1–10) for each image, reflecting their holistic impression. They also indicate whether the removal is **successful** or **unsuccessful** according to the same definition used in the AI evaluation—namely, that a successful result should appear seamless, contextually appropriate, and free of noticeable flaws.

Final results are reported in terms of AI and human perceptual scores, along with success rates computed as the percentage of cases marked as successful.

**Evaluation Results**   Table 8 summarizes the perceptual evaluation results from both AI and human annotators. Our method, **MILD**, consistently achieves the highest overall perceptual score among all compared approaches. It also attains the highest success rates from GPT-4o and from human

| Method | AI Overall | Human Overall | Success Rate |
|---|---|---|---|
| SDXL Inpaint | 3.7 | 4.2 | 53% / 50% |
| PowerPaint | 5.0 | 5.2 | 58% / 54% |
| Inst-Inpaint | 4.7 | 4.8 | 48% / 41% |
| LaMa | 6.4 | 6.3 | 67% / 63% |
| CLIPAway | 3.9 | 4.0 | 52% / 49% |
| RoRem | 5.4 | 5.4 | 64% / 61% |
| **MILD (Ours)** | **8.3** | **8.4** | **82% / 78%** |

Table 8: Perceptual evaluation summary on the MILD dataset. AI and human scores reflect overall perceptual quality (1–10 scale).

raters, indicating strong agreement across evaluation modalities. Overall, these results confirm the effectiveness of our MILD strategy in producing perceptually convincing and contextually coherent removal outputs, outperforming prior methods in both automated and human assessments.

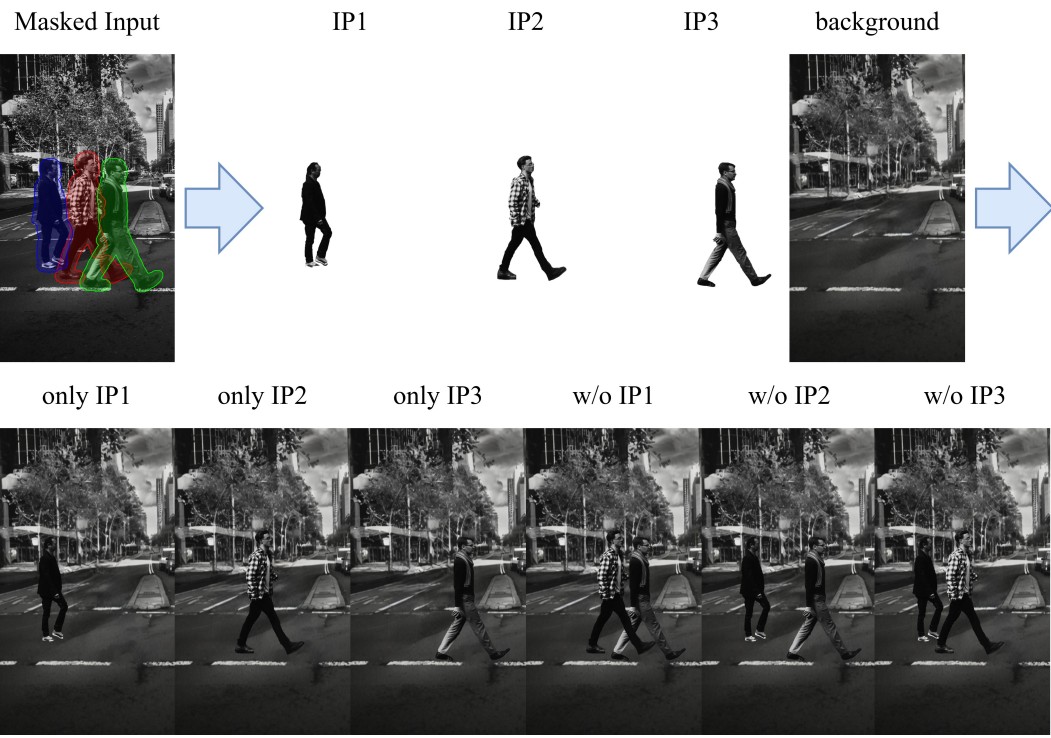

Figure 14: Compositional scene recomposition with MILD. Given a masked input with three foreground targets (IP1–IP3), we show the disentangled outputs and six representative recombinations enabled by our layered generation design.

## G.7 MORE COMPOSITIONAL FLEXIBILITY

Figure 14 provides a detailed visualization of MILD's compositional flexibility. Given a masked input image containing three interacting persons (IP1–IP3), our model generates four disentangled outputs: one foreground layer per instance and a clean background. By selectively combining these layers, MILD enables the construction of diverse scene variants. As illustrated, we showcase six recomposed outputs, including single-subject isolation (e.g., only IP1), targeted removal (e.g., w/o IP2), and partial restoration (e.g., IP1 and IP3). This layered formulation supports fine-grained, user-controllable editing, and highlights the advantages of explicit instance-level generation over unified inpainting approaches.

### G.8 ADDITIONAL STRATEGIES FOR ROBUST FACIAL PRESERVATION IN EXTREME CASES

To further strengthen MILD's robustness under extreme conditions,such as when the facial area of the target person is severely blurred, has low resolution, or is severely occluded, we introduce two optional lightweight enhancements. These strategies do not change MILD's core framework, but can be seamlessly integrated during inference to improve facial fidelity without sacrificing editing stability.

**Mask-based Isolation.** In multi-person scenes, we explicitly restrict the editable region to the labeled target individual by refining the human mask and excluding all faces of non-target individuals from the editable area. Pixels outside the refined mask are strictly preserved, and only the target human is processed by MILD. As illustrated in the left example of Figure 15, this spatial isolation effectively prevents unintended modifications to other people in the scene, ensuring their facial geometry and appearance remain faithful to the original input. This simple technique substantially reduces cross-person interference, especially when non-target individuals are located near the editing boundary.

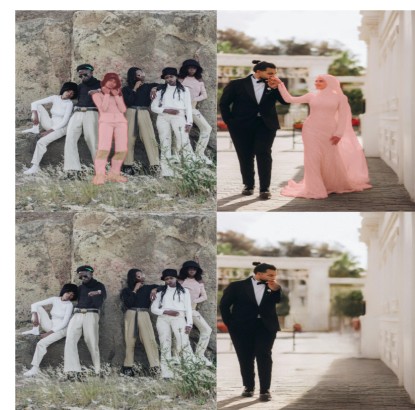

**Learnable Prompt Prefix.** Inspired by prompt-learning approaches such as CoOp Zhou et al. (2022), we introduce a small learnable prompt prefix that is optimized to reinforce identity-consistent reconstruction for heavily degraded facial regions. The prefix is trained on our dataset by conditioning the diffusion model on a compact set of learnable tokens, which guide the model toward preserving plausible human facial structures even when

Figure 15: Qualitative illustration of two inference-time strategies for mitigating facial distortion in non-target subjects.

the visual evidence is sparse or ambiguous. As shown in the right example of Figure 15, this learned prefix improves stability in extreme scenarios by reducing over-smoothing and artifact formation. Importantly, this enhancement is optional and lightweight: it does not require modifying model architecture, and it can be integrated at inference time without retraining the full model.

## H LIMITATIONS

Despite MILD's strong performance across challenging scenarios, several limitations remain. When the occluded area is extremely large (e.g., covering most of the image), MILD may still struggle to produce fully realistic completions, as contextual cues are insufficient for reconstructing fine background details (Figure 16, Row 1). In addition, the model can be overly conservative in some cases: although removal is successful and semantics remain coherent, it may under-restore background content or fail to fully reflect the underlying scene logic (Figure 16, Row 2). We view these challenges as promising directions for future work.

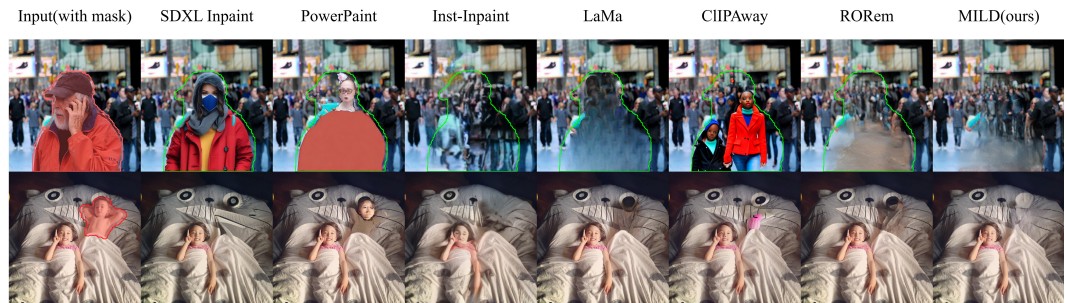

Figure 16: Failure example of MILD, along with other methods.

