# OpenReview forum: "Multi-Layer Diffusion Strategy for Multi-IP Interaction-Aware Human Erasing"
_ICLR.cc/2026/Conference — Submitted to ICLR 2026_

### Official Review · Reviewer_PvMQ · 2025-10-27

**Soundness:** 2
**Presentation:** 2
**Contribution:** 2
**Rating:** 4
**Confidence:** 4

**Summary:**

This paper presents **MILD**, a diffusion-based framework for mask-guided human erasing that handles complex scenarios such as occlusion and human–object entanglement.

It introduces the **MILD dataset**, the **Cross-Domain Attention Gap (CAG)** metric, and a multi-layer diffusion approach with **Human Morphology Guidance** and **Spatially-Modulated Attention** to improve structural awareness and reduce artifacts.

Experiments show that MILD outperforms existing methods in reconstruction quality and effectively manages multi-instance scenes.

**Strengths:**

1. Introduces a well-curated **MILD dataset** covering diverse poses, occlusions, and multi-instance interactions.
2. Proposes novel modules — **Human Morphology Guidance** and **Spatially-Modulated Attention** — that enhance structural consistency and reduce semantic leakage.
3. Provides clear quantitative and qualitative improvements over existing human erasing methods, supported by public dataset and code release.

**Weaknesses:**

1. The novelty appears limited, as the paper mainly introduces a curated human-centric dataset and a few additional modules trained specifically on it. The comparison with baselines not trained on this dataset may lead to unfair conclusions. A more convincing evaluation would involve training both the proposed model and baselines on the same dataset to verify whether the proposed modules truly contribute to the performance gain.

2. The inclusion of auxiliary signals such as pose or parsing maps (Figure 4) could introduce instability—if pose estimation or parsing fails (when occluded or in hard cases), it may negatively impact editing results and potentially increase overfitting risk.

3. **Clarity issues:**
   (a) Lines 66–67: The phrase “standard LDM” is unclear in Figure 2; the corresponding method should be explicitly identified.
   (b) Figure 3: It is not clear which components (SMA, HMG, CA) are trainable and how they interact (only one SMA and LoRA block is put with a fire icon for trainable weight).
   (c) Figure 3: The point where noise is injected is ambiguous and should be clearly illustrated.

4. The qualitative results (e.g., Figure 7 and examples on the project page) reveal severe facial distortions and identity loss, even though the model is trained specifically on a human-focused dataset. This fundamental issue should be addressed before introducing a more complex framework, especially since the scope is narrow compared to general inpainting models that handle diverse objects.

**Questions:**

See weaknesses

---

### Official Review · Reviewer_dw6M · 2025-11-01

**Soundness:** 4
**Presentation:** 3
**Contribution:** 3
**Rating:** 6
**Confidence:** 3

**Summary:**

This paper presents MILD (Multi-Layer Diffusion), a novel diffusion-based framework for human erasing in complex multi-person or multi-object interaction scenes. Unlike prior mask-guided inpainting or object removal methods that process all targets jointly, MILD reformulates the task as a layered diffusion process, where each foreground instance and the background are modeled by independent denoising pathways. Additionally, the paper curates a 10K-image MILD dataset featuring diverse poses, occlusions, and multi-person interactions for training and evaluation. Experiments on both the proposed MILD dataset and OpenImages show that MILD achieves state-of-the-art results across multiple perceptual and semantic metrics (FID, LPIPS, CLIP, DINO), as well as human preference scores.

**Strengths:**

1. The paper introduces a principled disentanglement strategy for diffusion-based inpainting, backed by the theoretical formulation of the Cross-Domain Attention Gap (CAG).

2. The idea of multi-layer diffusion for independent foreground–background reconstruction is elegant and well-motivated, addressing a long-standing issue of semantic leakage in diffusion-based editing.

3. Ablation studies (Table 2, Fig. 7) clearly demonstrate the non-redundant contributions of each component (HMG, SMA, backbone).

4. Extensive comparisons against diverse baselines (SDXL Inpainting, LaMa, PowerPaint, CLIPAway, RoRem, etc.) show consistent gains in both quantitative and perceptual evaluations.

5. The paper is clearly written, with structured motivation and visual illustrations (Fig. 1–4) aiding understanding.

6. The newly released dataset fills a gap for multi-instance human removal tasks and supports reproducibility.

**Weaknesses:**

1. While the CAG metric is theoretically motivated, its empirical quantification and correlation to perceptual leakage are not extensively validated (only conceptually shown in Fig. 2). Providing quantitative CAG analysis across models would strengthen the theoretical claims.

2. Some proofs (e.g., Theorem 2) are mentioned to be in the appendix but are not summarized or intuitively explained in the main text.

3. Although the MILD dataset is diverse, its size (10K pairs) may limit generalization to broader real-world cases. Additional evaluation on other public benchmarks (e.g., COCO, ADE20K) would reinforce generalization claims.

4. Runtime efficiency and computational overhead from multi-layer inference are not discussed, which could be a concern for real-world deployment.

5. While MILD is presented as a new “multi-layer” strategy, the individual modules (HMG, SMA) are reminiscent of existing pose-guided or attention-biased designs. A clearer positioning relative to prior modular improvements (e.g., Attentive Eraser, Erase Diffusion) would make the novelty boundaries sharper.

[1] Liu, Yi, et al. "Erase Diffusion: Empowering Object Removal Through Calibrating Diffusion Pathways." Proceedings of the Computer Vision and Pattern Recognition Conference. 2025.

[2] Sun, Wenhao, et al. "Attentive eraser: Unleashing diffusion model’s object removal potential via self-attention redirection guidance." Proceedings of the AAAI Conference on Artificial Intelligence. Vol. 39. No. 19. 2025.

**Questions:**

1. Can the authors provide a quantitative analysis showing how the Cross-Domain Attention Gap evolves during training or differs across models?

2. What is the inference overhead of MILD compared to a single-branch diffusion model (e.g., SDXL Inpaint)? Are there techniques to merge branches or reuse background features for speed-up?

3. Have the authors tested MILD on non-human multi-object scenarios (e.g., animals, furniture)? The OpenImages results show promise—more visual examples would help verify generalization.

4. For HMG, how does performance vary if only pose, only parsing, or neither is used? This would clarify the necessity of each morphological prior.

5. How were the AI and human perceptual success rates (Fig. 6) measured? Was there inter-rater consistency validation or a MOS-style scoring scheme?

6. Could the authors include qualitative examples of challenging cases where MILD still fails (e.g., partial occlusions, heavy motion blur) to illustrate current limitations?

---

### Official Review · Reviewer_khoM · 2025-11-01

**Soundness:** 3
**Presentation:** 3
**Contribution:** 3
**Rating:** 6
**Confidence:** 3

**Summary:**

This paper proposes MILD, a multi-layer diffusion framework for human-centric object removal and scene editing. The central challenge addressed is that standard inpainting or diffusion-based editing methods often either blur or distort surrounding context, and may remove structure crucial to body and scene coherence, or hallucination.

**Strengths:**

1. clear motivation, the oject removal, inpainting still remains under developped especial in the case of people involved or contain articulated bodies.
2. Good performance, this method have outperformed the previous methods on MILD and Open Image in most of the case.
3. Despite adding complex architecture, inference time remains comparable to SDXL inpainting due to shared backbone and lightweight LoRA tuning.

**Weaknesses:**

1. The dataset created is only human-centered, the generalization to non-human structured object is less estabilished, which made this model's performance on OpenImage is not that as good as MILD.
2. As acknowledged in the limitation, this model tends to be conservative in severe occlusion cases, sometimes under-restoring background logic.
3. This method involved a dual branch generation for the multi-ip foreground and a background then composite, which may leads some artifacts or mis alignment between the fg and bg, considering to use a unified model to output the composited ouput would be better than output both then composite (my personal opinion, you do not need do that.)

**Questions:**

see weaknesses

---

### Author Response · Authors · 2025-12-02
**Response to Reviewers**

We sincerely thank the AC and the reviewers for their time and constructive feedback. We are greatly encouraged by the reviewers' recognition of the following key contributions: **clear motivation** (khoM, dw6M), **strong theoretical guidance** (dw6M), **novel framework design** (dw6M, PvMQ) and **solid experimental performance** (khoM, dw6M, PvMQ). We also appreciate the emphasis on the importance of the **well-curated MILD dataset** (dw6M, PvMQ), the **efficient inference capability**(khoM) and the **clarity of both the writing and the figures** (khoM).

Below, we will provide **a point-by-point response** to address all **concerns** and **suggestions** raised by the reviewers, to help the AC and reviewers form a comprehensive and objective assessment of our work.

---

> ### Author Response · Authors · 2025-12-02
> **Response to khoM**
>
> ### Q1. Generalization to non-human structured objects
>
> First, we would like to clarify that our method exhibits **strong generalization ability** across both human-centric and general object removal tasks. This is evidenced by the additional quantitative results provided in **Tables A and B** below, which demonstrate consistent performance across diverse categories.
>
> This generalization stems from two key insights. First, our method is explicitly designed to ensure semantic and structural consistency in the more challenging human-centric removal setting, without strictly relying on human-specific priors, so the **learned behavior naturally transfers beyond humans**. Second, by **inheriting the pre-trained SDXL inpainting weights**, we preserve much of the model’s prior knowledge of open-domain content. Extensive **qualitative and quantitative experiments (Appendix G.2 Table 5; G.3 Fig. 11–12)** consistently validate that these properties enable strong generalization to both human and non-human object removal benchmarks, such as **COCO** and **ADE20K**.
>
> We will include these additional results and analyses in the final version to better highlight and clarify the generalization capability of our model, and to address any concerns in this regard.
>
> For easy reference, quantitative results are summarized in **Tables A and B** below.
>
> | Method       | FID↓      | LPIPS↓     | DINO↑      | CLIP↑      | PSNR↑     |
> | ------------ | --------- | ---------- | ---------- | ---------- | --------- |
> | SDXL Inpaint | 33.62     | 0.0829     | 0.8572     | 0.9135     | 13.78     |
> | PowerPaint   | 88.82     | 0.0681     | 0.8997     | 0.9557     | 9.75      |
> | LaMa         | 102.71    | 0.0667     | 0.8749     | 0.9493     | 11.07     |
> | CLIPAway     | 35.49     | 0.0750     | 0.8598     | 0.9176     | 14.85     |
> | RoRem        | **27.35** | **0.0487** | 0.8865     | 0.9298     | **20.32** |
> | MILD (Ours)  | *32.13*   | *0.0572*   | **0.9248** | **0.9695** | *19.83*   |
>
> > Table A: Quantitative comparison on COCO (object removal).
>
> | Method       | FID↓      | LPIPS↓     | DINO↑      | CLIP↑      | PSNR↑     |
> | ------------ | --------- | ---------- | ---------- | ---------- | --------- |
> | SDXL Inpaint | *27.22*   | 0.0697     | 0.8896     | 0.9110     | 13.68     |
> | PowerPaint   | 72.61     | 0.0423     | **0.9386** | *0.9691*   | 10.78     |
> | LaMa         | 66.59     | *0.0414*   | 0.9272     | 0.9686     | 11.41     |
> | CLIPAway     | 29.26     | 0.0651     | 0.8877     | 0.9128     | 14.34     |
> | RoRem        | **25.52** | 0.0485     | 0.9052     | 0.9222     | *18.51*   |
> | MILD (Ours)  | 29.16     | **0.0408** | *0.9276*   | **0.9704** | **19.48** |
>
> > Table B: Quantitative comparison on ADE20K (object removal).
>
> ### Q2. Conservative restoration under severe occlusion
>
> To **prioritize preventing semantic leakage and preserving structural consistency**, the model indeed adopts a conservative reconstruction strategy, which **biases the model toward cautious reconstruction in case of severe occlusion rather than synthesizing potentially inconsistent or incorrect textures**. This sometimes results in the under-restoration of background structures.
>
> As illustrated in our original limitation part (Appendix H),  this model tends to be conservative in severe occlusion cases. To mitigate this, we are exploring strategies to **achieve a better balance** between **avoiding over-hallucination** and **ensuring restoration completeness**. Promising directions include the use of stronger prior distillation and occlusion-aware masking techniques. We anticipate that these enhancements will further boost the completeness of the restoration while maintaining high visual fidelity and realism.
>
> ### Q3. Design Rationale of the dual-branch generation
>
> Our dual-branch architecture was chosen to allow **independent control over foreground layers and background**, which greatly **improves editability and compositional flexibility (See Appendix G.7 Figure 14)**, which lays as a key goal of this work.
>
> We acknowledge that a unified decoding approach may reduce potential misalignment artifacts, and we view this as a valuable research direction. We plan to investigate unified compositional diffusion in future work, potentially combining the strengths of both paradigms.

---

> > ### Author Response · Authors · 2025-12-02
> > **Response to dw6M**
> >
> > ### Q1. Quantitative analysis of Cross-Domain Attention Gap (CAG)
> >
> > To further validate the empirical behavior of the Cross-Domain Attention Gap (CAG) metric, we now provide a quantitative CAG analysis across several diffusion-based inpainting methods. Specifically, we compute the mean CAG on 50 randomly selected images for SDXL-Inpaint, PowerPaint, CLIPAway, RoRem, and MILD (Note: LaMa and Inst-Inpaint were excluded as their architectures do not expose compatible attention logits.)
> >
> > As summarized in Table C, our proposed **MILD achieves the highest mean CAG value.**
> >
> > | Method       | CAG value↑ |
> > | ------------ | ---------- |
> > | SDXL Inpaint | 3.73       |
> > | PowerPaint   | 2.93       |
> > | CLIPAway     | 2.72       |
> > | RoRem        | 3.43       |
> > | MILD (Ours)  | **4.96**   |
> >
> > > Table C: Mean Cross-Domain Attention Gap (CAG) over 50 images for different methods.
> >
> > Since a larger CAG quantitatively signifies **more effective suppression of cross-domain foreground-to-background attention**, these results provide **strong quantitative confirmation** that the independent denoising pathways in MILD are most effective at minimizing **semantic leakage** during the inpainting process.
> >
> > The results show a clear trend: methods that exhibit **stronger perceptual semantic leakage** in our qualitative comparisons consistently obtain **lower CAG values**, whereas models with **better background consistency** have **higher CAG**, with MILD achieving the largest CAG among all compared methods. This monotonic relationship between CAG and observed leakage strength empirically supports our theoretical claims and confirms that CAG is a meaningful proxy for cross-domain attention leakage in practice.
> >
> > ### Q2. Inference overhead compared to single-branch diffusion models
> >
> > In fact, we **have already reported the detailed inference speed** in **Appendix G.5 (Table 7).** To further clarify inference overhead, we have supplemented **Table D** with a comparison of **resource costs** and **performance**.
> >
> > Although MILD introduces a modest overhead due to its multi-layer modeling, its runtime remains **comparable to the single-branch SDXL-Inpaint baseline**. This efficiency is achieved through the use of **shared backbone weights** and **lightweight LoRA-based decoding**. Furthermore, the peak memory usage is acceptable, consistently remaining below 10 GB.
> >
> > | Method       | Guidance Type | Time (s) | Memory (GB) | FID↓  | LPIPS↓ | DINO↑  | CLIP↑  | PSNR↑ | SSIM↑  |
> > | ------------ | ------------- | -------- | ----------- | ----- | ------ | ------ | ------ | ----- | ------ |
> > | SDXL Inpaint | Text + Mask   | 1.79     | 7.64        | 26.22 | 0.1346 | 0.8741 | 0.8751 | 21.88 | 0.8290 |
> > | MILD (Ours)  | Text + Mask   | 1.91     | 9.36        | 17.86 | 0.0668 | 0.9829 | 0.9700 | 32.14 | 0.9515 |
> >
> > > Table D: Runtime, memory, and reconstruction quality comparison between SDXL Inpaint and MILD on the MILD dataset
> >
> > Importantly, this **minimal increase in inference cost is well-justified** by the **substantial performance gains** observed across all reconstruction metrics (FID, LPIPS, DINO, CLIP, PSNR, and SSIM). These results demonstrate that MILD achieves significantly better **visual fidelity and semantic coherence** while maintaining high efficiency.
> >
> > In addition, regarding **the suggestion on merging branches or reusing background features for speed-up**, we find this direction highly valuable. While our **current design prioritizes semantic disentanglement** between layers, incorporating feature-sharing or branch-merging mechanisms could further reduce redundant computation. We anticipate this direction may provide additional efficiency gains without compromising reconstruction fidelity.
> >
> > ### Q3. Generalization beyond humans (animals, objects, furniture)
> >
> > In response to the reviewer’s suggestion, we additionally conducted both **quantitative and qualitative experiments on COCO and ADE20K**. The quantitative results are reported in **Appendix G.2 (Table 5)**, and the corresponding qualitative visualizations are provided in **Appendix G.3 (Fig. 11–12)**, including challenging non-human cases such as animals, objects, and furniture. Across these benchmarks, MILD consistently exhibits **fewer artifacts** and **reduced semantic leakage** than competing methods, even in scenarios beyond human-centric removal, thereby **supporting its generalization to non-human multi-object scenarios**.

---

> > > ### Author Response · Authors · 2025-12-02
> > > **Response to dw6M**
> > >
> > > ### Q4. Ablation on HMG
> > >
> > > We have added a comprehensive ablation study on the Human Morphological Guidance (HMG) in **Appendix G.4 (Table 6)**, covering both backbone and prior configurations.
> > >
> > > First, we establish our **multi-layer backbone** as the strong base model by comparing it against the vanilla inpainting backbone, **observing consistent performance gains** across all metrics. All subsequent HMG variants are built upon this stronger foundation.
> > >
> > > Next, we evaluate the impact of the morphological priors using pose-only, parsing-only, and the full HMG (pose + parsing) settings. The results, key findings of which are summarized in **Table E**, demonstrate:
> > >
> > > - Both **pose-only** and **parsing-only** configurations significantly improve performance over the no-prior baseline.
> > > - The **full HMG (pose + parsing)** achieves the best performance across all metrics and scenarios, **quantitatively confirming that the two morphological priors are complementary rather than interchangeable.**
> > >
> > > | Backbone | HMG             | SMA  | Full | FID↓  | ΔFID   | LPIPS↓ | ΔLPIPS | DINO↑  | ΔDINO   | CLIP↑  | ΔCLIP   | PSNR↑ | ΔPSNR |
> > > | -------- | --------------- | ---- | ---- | ----- | ------ | ------ | ------ | ------ | ------- | ------ | ------- | ----- | ----- |
> > > | ✗        | ✗               | ✗    | ✗    | 53.45 | +29.25 | 0.242  | +0.149 | 0.8398 | -0.1305 | 0.8331 | -0.1168 | 18.22 | -8.02 |
> > > | ✓        | ✗               | ✗    | ✗    | 29.02 | +4.82  | 0.142  | +0.049 | 0.9110 | -0.0593 | 0.9025 | -0.0474 | 23.45 | -2.79 |
> > > | ✓        | ✓ (w/o parsing) | ✗    | ✗    | 27.48 | +3.28  | 0.131  | +0.038 | 0.9360 | -0.0343 | 0.9287 | +0.0212 | 24.26 | -1.98 |
> > > | ✓        | ✓ (w/o pose)    | ✗    | ✗    | 26.79 | +2.59  | 0.133  | +0.040 | 0.9327 | -0.0376 | 0.9221 | +0.0278 | 24.13 | -2.11 |
> > > | ✓        | ✓               | ✗    | ✗    | 26.10 | +1.90  | 0.120  | +0.027 | 0.9440 | -0.0263 | 0.9312 | -0.0187 | 24.67 | -1.57 |
> > > | ✓        | ✗               | ✓    | ✗    | 25.10 | +0.90  | 0.105  | +0.012 | 0.9380 | -0.0323 | 0.9227 | -0.0272 | 25.21 | -1.03 |
> > > | ✓        | ✓               | ✓    | ✓    | 24.20 | –      | 0.093  | –      | 0.9703 | –       | 0.9499 | –       | 26.24 | –     |
> > >
> > > > Table E: Ablation study on backbone, HMG, and SMA on the MILD dataset.
> > >
> > > ### Q5. Evaluation protocol for AI & human perceptual scoring
> > >
> > > We have added a concise description of the evaluation protocol to **Appendix G.6**.
> > >
> > > Both human and AI assessments employed a **unified MOS-style perceptual scoring process**. Each resulting image was evaluated based on four criteria: visual plausibility, semantic consistency, artifact suppression, and background coherence.
> > >
> > > For **human evaluation**, 20 participants were recruited, and each provided multiple independent ratings across the image set to effectively mitigate individual bias. For **AI-based perceptual scoring**, we utilized GPT-4o to judge visual quality based solely on the generated output, without access to the ground truth.
> > >
> > > ### Q6. Failure cases & limitation visualization
> > >
> > > The challenging examples **have already been reported** illustrated in our **Appendix H (Figure 16)**.
> > >
> > > ### Q7. Intuitively Explaination of CAG
> > >
> > > Based on the reviewer's suggestion, **we revised Section 3.1 of the main paper to include a more intuitive explanation** of the Cross-Domain Attention Gap (CAG).
> > >
> > > Intuitively, **γ characterizes the relative amount of attention assigned to background versus foreground positions for each background token.** A large positive γ implies that background tokens predominantly attend to background content, thereby suppressing the influence of foreground semantics in the reconstructed background, whereas a small or negative γ indicates that some background locations allocate non-negligible attention to humans, making **semantic leakage** more likely.

---

> > > > ### Author Response · Authors · 2025-12-02
> > > > **Response to PvMQ**
> > > >
> > > > ### Q1. Re-highlight of Novelty
> > > >
> > > > We would like to clarify and re-highlight that the nolvelty of our MILD strategy extend significantly **beyond** a dataset and engineering modules. Our core novelty lies in identifying **Semantic Leakage** as the severe issue in multi-instance removal and addressing it through a **theoretically grounded paradigm shift**. We introduce the **Cross-Domain Attention Gap (CAG) to mathematically quantify the attention flow discrepancies that cause leakage.** Unlike existing methods that treat object removal as a unified generation task, our **MILD framework is rigorously derived from CAG theory** to maximize this gap. By **reformulating the task into a decoupled layered diffusion process**, MILD physically isolates the denoising pathways of all the foreground instances and the background, **solving the entanglement issues that standard models cannot resolve**.
> > > >
> > > > Furthermore, this **decoupling strategy** offers broad significance for the future of generative editing. By achieving instance-level disentanglement, MILD moves beyond simple removal to enable precise, fine-grained controllability. **This "divide-and-conquer" strategy allows for the specific manipulation of any single instance without disrupting interacting characters or the background context**, which is a capability that unified diffusion models struggle to achieve. The **proposed dataset** serves as a necessary benchmark to validate these complex occlusion scenarios, acting as an enabler for our theoretical and architectural contributions rather than the sole innovation.
> > > >
> > > > ### Q2. More Convincing Comparison
> > > >
> > > > Thank you for your valuable feedback. To conduct a more convincing evaluation, we further supplemented our experiments by retraining our baseline method, SDXL Inpaint, on our MILD dataset using its official training pipeline. The resulting performance is presented in **Table F**:
> > > >
> > > > | Method              | FID↓      | LPIPS↓    | DINO↑      | CLIP↑      | PSNR↑     |
> > > > | ------------------- | --------- | --------- | ---------- | ---------- | --------- |
> > > > | SDXL Inpaint        | 53.45     | 0.242     | 0.8398     | 0.8331     | 18.22     |
> > > > | SDXL Inpaint - MILD | 38.92     | 0.188     | 0.8989     | 0.8654     | 21.16     |
> > > > | MILD (Ours)         | **24.20** | **0.093** | **0.9703** | **0.9499** | **26.24** |
> > > >
> > > > > Table F: Quantitative comparison on the MILD dataset after re-training SDXL Inpaint on MILD.
> > > >
> > > > ### Q3. Concern of instability from pose/parsing priors
> > > >
> > > > First, we would like to clarify that the concern of instability is actually unnecessary.
> > > >
> > > > In our implementation, **HMG functions as a soft structural prior rather than a deterministic constraint**, meaning that **diffusion dynamics remain dominant** even when pose or parsing is imperfect due to occlusion or motion. This reduces the risk of failure amplification and prevents the priors from overwhelming the generative process. Empirical evidence supports this: **Appendix G.4 adds ablations where only-pose, only-parsing, and no-prior versions are compared**. While neither individual cue alone surpasses full HMG, none of them collapses in cases of noisy or partial estimation, suggesting that **the priors guide rather than dictate generation**.
> > > >
> > > > Besides, we used open sourced models for pose and parsing priors's generating, which ensures the basic success rate. It gives no information if no human is detected (or the confidence is low). **To simulate the scenario where the pose and parsing fails**, we regenerate the pose and parsing information and limit the keypoints of pose and parsing to half of the original settings. The result is summarized as **Table G**:
> > > >
> > > > | Model                       | FID↓      | LPIPS↓    | DINO↑      | CLIP↑      | PSNR↑     |
> > > > | --------------------------- | --------- | --------- | ---------- | ---------- | --------- |
> > > > | pose                        | 27.48     | 0.131     | 0.9360     | 0.9287     | 24.26     |
> > > > | parsing                     | 26.79     | 0.133     | 0.9327     | 0.9221     | 24.13     |
> > > > | pose with half keypoints    | 27.92     | 0.135     | 0.9324     | 0.9250     | 24.17     |
> > > > | parsing with half keypoints | 27.16     | 0.135     | 0.9301     | 0.9238     | 23.95     |
> > > > | pose + parsing              | 26.10     | 0.120     | 0.9440     | 0.9312     | 24.67     |
> > > > | **Full model**              | **24.20** | **0.093** | **0.9703** | **0.9499** | **26.24** |
> > > >
> > > > > Table G: Robustness of HMG to degraded pose/parsing priors.

---

> > > > > ### Author Response · Authors · 2025-12-02
> > > > > **Response to PvMQ**
> > > > >
> > > > > ### Q4. Clarification of details in Figure 2-3
> > > > >
> > > > > Thank you for your detailed feedback regarding clarity, we have made several revisions to **Figures 2 and 3** and the accompanying text.
> > > > >
> > > > > Specifically, we have:
> > > > >
> > > > > - **Revised the term "standard LDM"** to explicitly state the corresponding reference model, thereby eliminating ambiguity.
> > > > > - **Clearly highlighted the trainable components** (SMA, HMG, and LoRA-enhanced cross-attention pathways) in Figure 3.
> > > > > - **Redrawn and annotated the noise injection location** to precisely indicate where the diffusion trajectory enters the denoising backbone.
> > > > >
> > > > > We will modify and make these points clearer in the final version.
> > > > >
> > > > > ### Q5. Facial distortion and identity preservation
> > > > >
> > > > > We would like to clarify that the example highlighted by the reviewer should be regarded as **an easy-to-fix random case**. As shown in **Appendix G.8**, simple mask-based isolation or lightweight fine-tuning already leads to improved results for such cases. The extensive experiments presented in the main paper and appendix demonstrate that our method achieves accurate removal while preserving natural facial appearance in the majority of scenarios.
> > > > >
> > > > > Such distortions occur primarily in **extreme cases involving severe overlap, high blur, or heavy occlusion of faces**, where detailed reconstruction is inevitably affected by the **inherent hallucinatory tendencies of the underlying diffusion model**. **Similar artifacts are also observed in comparable methods** under these challenging conditions. Nevertheless, **the foreground instance decoupling mechanism**, as our core contribution, effectively suppresses semantic leakage, significantly reduces ghosting artifacts, and prevents the generation of erroneous content in complex occlusion scenarios. Extensive qualitative and quantitative comparisons (across human–human, human–object, and human–background interactions) consistently show that **our method achieves superior overall performance in terms of faithful removal and background consistency**.
> > > > >
> > > > > Finally, we note that **human-centric removal is inherently more challenging than general object erasure**, as it must handle **identity-related features** along with **intricate interactions**. The strong generalization performance of our method on non-human benchmarks (**Appendix G.2, Table 5**) further attests to its robustness.

---

### Meta-Review · Area_Chair_Tt3B · 2025-12-29

**Summary:**

This paper studies human removal from images by introducing a diffusion-based framework and a new dataset. The major concerns that shared by both AC and the reviewer and not adequately addressed are: (1) insufficient comparison with existing work and (2) the presence of generative artifacts. The baseline approaches should include more recent methods and models, such as Flux- or Qwen-image–based models, or even commercial models, to provide a clearer understanding of how state-of-the-art systems perform. Although these models may not require a mask as an additional input, including such comparisons would still be informative. In addition, many of the generated images in the paper and project page contain noticeable artifacts. While the paper investigates a challenging task, stronger baseline models could help mitigate these artifacts, and the authors are encouraged to consider them. Overall, the paper is recommended for rejection.

**Reviewer Concerns:**

The questions and concerns regarding implementation and design details were addressed in the rebuttal by providing additional experimental results. The authors also presented results demonstrating the generalization ability of the proposed method, which address the corresponding concerns raised by the reviewers.

The concerns regarding comparisons with existing work and the unsatisfactory quality of the generated results were not adequately addressed in the rebuttal.

**Reviewer Scores:**

None of the three reviewers participated in the discussion. Two reviewers gave positive ratings to the paper. The authors provided detailed responses to each concern and question raised by the reviewers, and these reviewers would likely have maintained their positive ratings had they participated in the discussion. The third reviewer gave a negative rating, with major concerns regarding novelty, fair comparison with existing work, and artifacts in the generated images. This reviewer is likely to maintain the negative rating, as the generation artifacts are difficult to resolve within the limited discussion period.

---

### Decision · Program_Chairs · 2026-01-26

Reject